# Inflammation drives alternative first exon usage to regulate immune genes including a novel iron-regulated isoform of *Aim2*

**Elektra K Robinson[1†], Pratibha Jagannatha[1,2†], Sergio Covarrubias[1], Matthew Cattle[2], Valeriya Smaliy[1], Rojin Safavi[2], Barbara Shapleigh[1], Robin Abu-Shumays[2], Miten Jain[2], Suzanne M Cloonan[3], Mark Akeson[2], Angela N Brooks[2*], Susan Carpenter[1*]**

[1]Department of Molecular, Cell and Developmental Biology, University of California Santa Cruz, Santa Cruz, United States; [2]Department of Biomolecular Engineering, University of California Santa Cruz, Santa Cruz, United States; [3]Division of Pulmonary and Critical Care Medicine, Joan and Sanford I. Weill Department of Medicine, Weill Cornell Medicine, New York, United States

**\*For correspondence:**
anbrooks@ucsc.edu (ANB);
sucarpen@ucsc.edu (SC)

†These authors contributed equally to this work

**Abstract:** Determining the layers of gene regulation within the innate immune response is critical to our understanding of the cellular responses to infection and dysregulation in disease. We identified a conserved mechanism of gene regulation in human and mouse via changes in alternative first exon (AFE) usage following inflammation, resulting in changes to the isoforms produced. Of these AFE events, we identified 95 unannotated transcription start sites in mice using a de novo transcriptome generated by long-read native RNA-sequencing, one of which is in the cytosolic receptor for dsDNA and known inflammatory inducible gene, *Aim2*. We show that this unannotated AFE isoform of *Aim2* is the predominant isoform expressed during inflammation and contains an iron-responsive element in its 5'UTR enabling mRNA translation to be regulated by iron levels. This work highlights the importance of examining alternative isoform changes and translational regulation in the innate immune response and uncovers novel regulatory mechanisms of *Aim2*.

## Introduction

Macrophages are critical cells in the innate immune system that combat infection by initiating acute inflammatory responses. Acute inflammation is tightly coordinated and begins with the detection of pathogen-associated molecular patterns (PAMPs) by pattern recognition receptors (PRRs), which include toll-ike receptors (TLRs) (*Medzhitov and Janeway, 1998*; *Pai et al., 2016*). These initial steps are followed by the activation of sequestered transcription factors (TFs), such as nuclear factor of kappa-B (NF-κB) and interferon regulatory factors (IRFs), which orchestrate pro-inflammatory and antiviral response signals involved in pathogen clearance (*Pai et al., 2016*). Once pathogens are cleared, macrophages express genes involved in the resolution of inflammation to return the host to homeostasis (*Hamidzadeh et al., 2017*). Dysregulation of these pro-inflammatory pathways can have devastating consequences, leading to unresolved inflammation and chronic inflammatory diseases (*Zhou et al., 2016*).

Recently, the process of alternative splicing has emerged as another key mechanism by which the immune system is regulated. Alternative splicing is a regulated process enabling a single gene to produce many isoforms, thus increasing the complexity of gene function and the proteome

(*Boudreault et al., 2016*; *Ivanov and Anderson, 2013*; *Pai et al., 2016*; *Wang et al., 2015*). Much of this occurs in a cell-type-specific and signal-induced manner (*Ergun et al., 2013*; *Wells et al., 2006*). Previous studies have shown that mouse and human macrophages exposed to inflammatory stimuli undergo alternative splicing (*Beyer et al., 2012*; *Bhatt et al., 2012*; *de Bruin et al., 2016*; *Haque et al., 2018*; *Janssen et al., 2020*; *Lin et al., 2016*; *Liu et al., 2018*; *O'Connor et al., 2015*; *Pai et al., 2016*; *Pandya-Jones et al., 2013*). Alternative splicing within the immune system can affect the type and magnitude of the inflammatory response, such as the production of a soluble form of TLR4 that is expressed upon lipopolysaccharide (LPS), which leads to inhibition of TNFα and NF-κB serving as a negative feedback mechanism (*Lynch, 2004*; *Schaub and Glasmacher, 2017*). Additionally, this mechanism has been characterized within signaling molecules (*Blumhagen et al., 2017*; *Shakola et al., 2015*), including TBK1 (*Deng et al., 2008*) and MyD88 (*De Arras and Alper, 2013*), that produce the alternative RNA splice forms, TBK1s and MyD88s, respectively, which function to limit the extent of the pro-inflammatory response. Alternative splicing can also result in the production of inflammatory signaling molecules, such as TRIF (*Han et al., 2010*) and the proteins in the NFκB family (*Wells et al., 2006*) with altered activity or stability. Beyond changing the ORF of an mRNA molecule, elongating or shortening the first or last exon can impact post-transcriptional gene regulation and are important to consider when elucidating the regulatory mechanisms of immune genes (*Carpenter et al., 2014*; *Ghiasvand et al., 2014*; *Leppek et al., 2018*), specifically underlying motifs in 5′ untranslated regions (UTRs) (*Kramer et al., 2013*; *Resch et al., 2009*) and 3′UTRs (*Mariella et al., 2019*; *Mayr, 2016*).

While inflammation-induced alternative splicing in both human and mouse macrophages has been investigated on a genome-wide scale (*Beyer et al., 2012*; *Bhatt et al., 2012*; *de Bruin et al., 2016*; *Haque et al., 2018*; *Janssen et al., 2020*; *Lin et al., 2016*; *Liu et al., 2018*; *O'Connor et al., 2015*; *Pai et al., 2016*; *Pandya-Jones et al., 2013*), to our knowledge, long-read RNA-sequencing has not been utilized to generate a de novo transcriptome for primary murine macrophages. Such an approach is necessary to fully appreciate the extent of alternative transcript isoform usage as we know most transcriptome annotations are incomplete (*Workman et al., 2019*) and isoforms generated are cell-type and treatment specific (*Sapkota et al., 2019*; *Weirather et al., 2017*; *Workman et al., 2019*; *Wu et al., 2018*).

Here we used both long- and short-read RNA-sequencing to uncover novel isoforms and classes of alternative splicing events following inflammation in human and murine macrophages. Interestingly the dominant conserved class of alternative isoform usage observed following inflammation is alternative first exon (AFE) usage, which involves alternative transcription start sites (TSS) usage coupled with alternative splicing. AFE events can have multilevel effects on protein diversity, regulating genes through alterations of the 5′UTR region, and directing the locality of proteins through alternative N-termini (*Landry et al., 2003*). We identified 95 unannotated AFE events in mice from native RNA-sequencing, one of which is in the cytosolic receptor for dsDNA and known inflammatory inducible gene, *Aim2*. We show that this unannotated AFE isoform of *Aim2* is the predominant isoform produced during inflammation and contains an iron-responsive element (IRE) in its 5′UTR, enabling mRNA translation to be controlled by iron levels. This work reveals that alternative transcript isoform usage plays a crucial role in shaping the transient nature of the inflammatory response. Isoform expression is an additional layer of regulation within the immune response and therefore a possible contributing factor to the development of auto-immune and inflammatory diseases. Understanding the exact isoforms of genes that are expressed during an inflammatory response will enable us to design better targets for therapeutic intervention of these diseases.

## Results

### Global profiling of the cellular alternative splicing landscape in human and mouse macrophages post-inflammation

To identify alternative splicing events following inflammation, we performed whole transcriptome analysis on human monocyte-derived macrophages (MDMs) and murine bone marrow-derived macrophages (BMDMs) with and without LPS treatment (*Figure 1A*). We found that ~50% of splicing changes (corrected p-value ≤ 0.25 and |ΔPSI| ≥ 10) were classified as AFE events following LPS activation in both human and murine macrophages (*Figure 1B*, *Figure 1—figure supplement 1*). A ranking analysis of the significant events, from both mouse and human data sets, revealed that AFE events

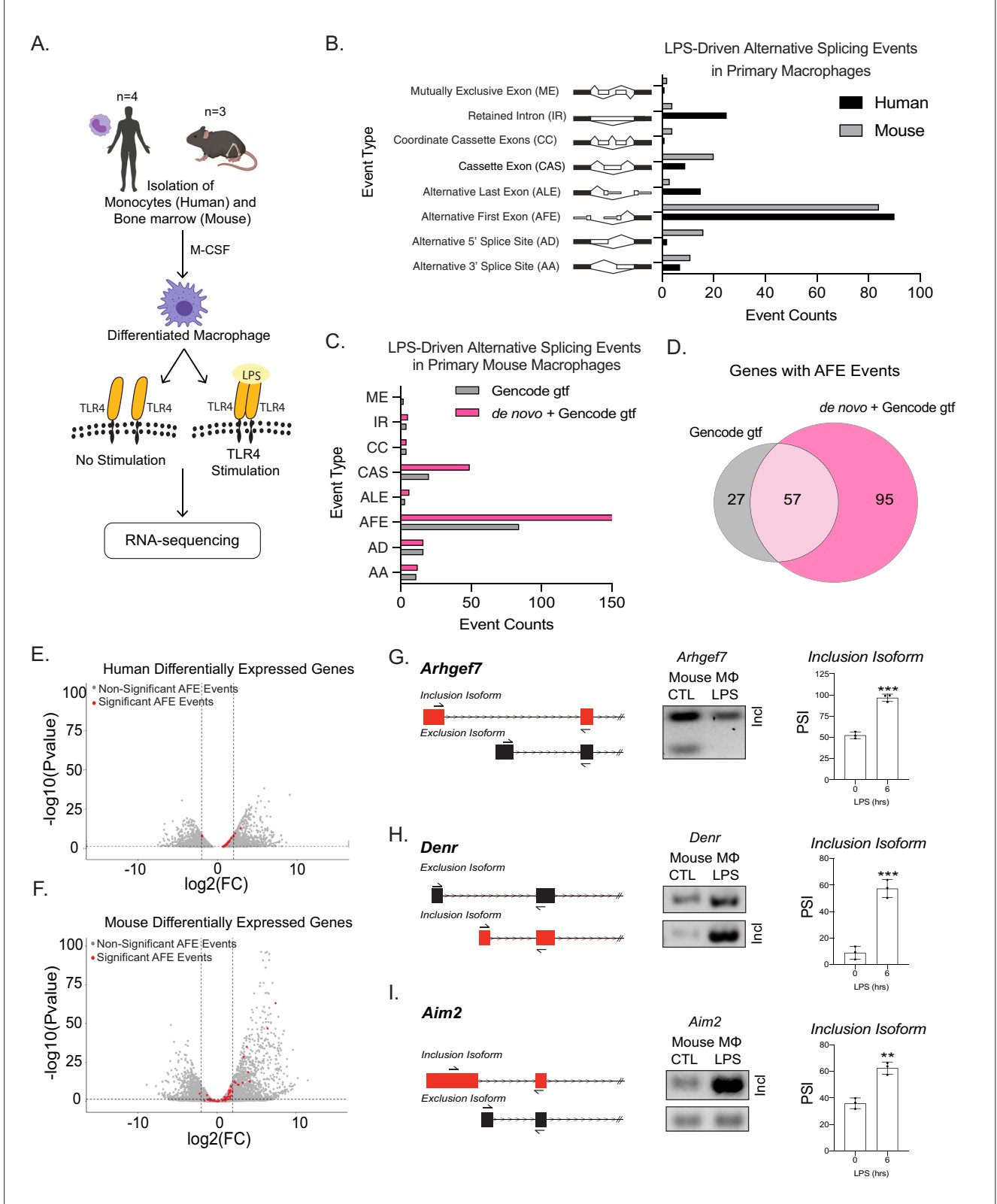

**Figure 1.** Global profiling of the cellular alternative splicing landscape in human and mouse macrophages post-inflammatory. (**A**) Diagram of RNA-seq library generation. (**B**) Categorization of significant splicing events in human and mouse macrophage. (**C**) Categorization of significant splicing events found in mouse bone marrow-derived macrophages (BMDM) ±6 hr lipopolysaccharide by using either the Gencode annotation or the GENCODE+ de novo annotation. (**D**) Venn diagram representing unique and common genes with alternative first exon (AFE) events found in RNA-seq of primary

*Figure 1 continued on next page*

*Figure 1 continued*

BMDMs post-inflammatory stimulation using the Gencode annotation or the GENCODE+ de novo annotation. Volcano plots of all differentially expressed genes from RNA-seq of either human (**E**) or mouse (**F**) macrophages. Genes highlighted in red undergo significant AFE changes following inflammation. Schematic of AFE inclusion and exclusion isoforms, followed by RT-PCR gel results and percent spliced in calculation for *Argehf7* (**G**) *Denr* (**H**), and *Aim2* (**I**), was performed in biological triplicates, p-value assessed using Student's t-test.

The online version of this article includes the following figure supplement(s) for figure 1:

**Figure supplement 1.** Computational pipeline and comparison of t-test and DRIMSeq alternative splicing events.

**Figure supplement 2.** Alternative first exon events remain prevalent amongst ranked alternative splicing events in human macrophages.

**Figure supplement 3.** Alternative first exon events remain prevalent amongst ranked alternative splicing events in mouse macrophages.

**Figure supplement 4.** Alternative first exon usage is the top splicing event when using JuncBASE to identify and quantify alternative splicing events from primary human macrophages stimulated with *Listeria* and *Salmonella* for 24 hr.

**Figure supplement 5.** Conserved genes identified through alternative splicing events between human and mouse using gencode transcriptome.

**Figure supplement 6.** Validation of mouse and human alternative first exon events.

**Figure supplement 7.** Comparison of alternative splicing events identified using the gencode transcriptome or the GENCODE+ de novo transcriptome.

**Figure supplement 8.** CAGE scores support validity of novel transcription start sites (TSS) identified by nanopore sequencing.

**Figure supplement 9.** Validated unannotated alternative first exon (AFE) isoforms.

---

consistently comprised a large proportion of the top splicing changes (***Figure 1—figure supplements 2 and 3***). Additionally, analysis of previously published primary human macrophages stimulated with either *Listeria* or *Salmonella*, using our bioinformatic pipeline and with more stringent thresholds than the previous study, also revealed AFE events to be amongst the most significant prevalent alternative splicing events (corrected p-value ≤ 0.05 or 0.25 and |ΔPSI| ≥ 10) (***Figure 1—figure supplement 4***; ***Supplementary files 4-5–***; ***Pai et al., 2016***).

We next identified 11 conserved AFE splicing events between human and mouse (***Figure 1—figure supplement 5A,B***), the largest number of conserved event types amongst all alternative splicing event types (***Figure 1—figure supplement 5***). We validated the AFE changes upon stimulation on the already characterized *Ncoa7* (***Figure 1—figure supplement 6A–C***; ***Singer et al., 2008***; ***Yu et al., 2015***) and *Rcan1* (***Figure 1—figure supplement 6D–F***; ***Pang et al., 2018***), as well as a previously uncharacterized inflammatory-specific isoform of *Ampd3* (***Figure 1—figure supplement 6G–I***), in human and mouse primary macrophages using RT-PCR. Taken together, these results show the high prevalence and conservation of AFE usage following inflammatory activation.

A caveat to our analysis thus far was the reliance on annotated transcriptome assemblies to identify first exons of genes (***Brooks et al., 2015***). In order to determine if there are additional splicing events that are not captured using the publicly available GENCODE M18 transcriptome annotation (***Garalde et al., 2018***; ***Hsieh et al., 2019***; ***Pollard et al., 2018***; ***Workman et al., 2019***), we performed native RNA-sequencing of murine macrophages with and without LPS treatment to build a de novo murine macrophage specific transcriptome with an average read depth of 1 million. We identified isoforms using full-length alternative isoform analysis of RNA (FLAIR) (***Tang et al., 2020***; ***Workman et al., 2019***) that also had promoter support identified from accessible chromatin (ATAC-seq) (***Atianand et al., 2016***; ***Tong et al., 2016***). The FLAIR isoforms were then merged with the GENCODE M18 assembly (mm10) (***Frankish et al., 2019***) as a transcript reference to identify and quantify alternative splicing events using short-read sequencing, which has increased read depth, denoting this transcriptome as 'GENCODE+ de novo' (***Figure 1—figure supplement 7A***). Overall, the incorporation of long-read sequencing to generate a novel transcriptome led to the identification of 95 novel and statistically significant AFE events that occur following inflammation (***Figure 1C,D***). A comparison of significant AS events between splicing events identified by using GENCODE M18 annotation and de novo FLAIR transcriptome shows an overall increase in the number of unique events (***Figure 1—figure supplement 7B,H***). To gain confidence of the novel AFE events, we compared a comprehensive, 339 mouse sample, FANTOM CAGE-seq data (***FANTOM Consortium and the RIKEN PMI and CLST (DGT), 2014***) or annotated and unannotated TSSs as defined by the GENCODE M18 assembly. The caveat to this FANTOM data set is that there are no inflammatory stimulated samples. Even with this caveat, the analysis revealed the similarity of the distributions between CAGE scores associated with CAGE peaks overlapping known and those overlapping novel TSSs of FLAIR isoforms, further supporting the

validity of the novel TSSs identified (*Figure 1—figure supplement 8*, *Figure 3—figure supplement 1B*). Additionally, we utilized a data set generated through application of machine learning methods to classify CAGE-seq peaks as true or false, where true corresponds to CAGE-seq peaks that overlap true TSSs (*Kanamori-Katayama et al., 2011*). Novel TSSs identified with FLAIR showed a 45% overlap with CAGE-seq peaks classified as true, providing us with confidence in the novel AFE events identified (*Figure 1—figure supplement 8C*). Together these approaches provide us with confidence in the novel AFE events identified by direct RNA-seq.

Interestingly, when identifying gene expression changes, we found that ~50% of genes with AFE usage were not differentially expressed following inflammation (*Figure 1E,F*; *Supplementary files 6–7*), highlighting the importance of studying isoform usage for control of gene expression. Among the most statistically significant novel AFE first exon events were *Denr*, *Arhgef7*, and *Aim2*, which we validated using RT-qPCR (*Figure 1G–I*, *Figure 1—figure supplement 9*).

## Identification of an unannotated promoter for *Aim2*

To better understand the potential functional consequence of AFE changes, we further examined the novel first exon event upregulated upon inflammatory activation in *Aim2*. *Aim2* is an interferon-stimulated gene (ISG), localized to the cytosol. *Aim2* is a dsDNA sensor that upon recognition induces the formation of an inflammasome complex releasing IL1β and IL18 from the cell as a defense mechanism to control infection (*Wang and Yin, 2017*). Chromatin immunoprecipitation (ChIP)-seq for the myeloid pioneering TF PU.1 in primary BMDMs (*Figure 2A,B*, top track, in black) (*Lam et al., 2013*) supported the presence of an additional promoter upstream of the canonical isoform for *Aim2* (NM_001013779.2). Predominant isoforms (≥10% of total gene expression in a sample) assembled from native RNA-sequencing with FLAIR identified the canonical isoform and five unannotated isoforms that use the inflammatory-activated promoter, revealing a new longer 5′UTR (*Figure 2C*). Native RNA-sequencing-based quantification provided additional support that the unannotated promoter usage is upregulated upon LPS stimulation. At steady state, approximately 20% of reads map to *Aim2*'s transcript with the canonical promoter and 65% of reads map to transcripts with the upstream promoter, while following inflammatory activation, 14% of the reads map to transcripts with the canonical promoter and 81% of reads map to transcripts with the upstream promoter (*Figure 2D*). To validate the change in *Aim2* AFE usage upon LPS stimulation, RT-qPCR was performed using exon spanning primers that were either specific to the annotated AFE or the unannotated AFE, in BMDMs (*Figure 2E*). The expression profile of the annotated first exon is not induced by LPS stimulation (*Figure 2F*), while the unannotated first exon and the CDS of *Aim2* are equally induced by LPS stimulation (*Figure 2G,H*). Therefore, these data show that it is the novel isoform of *Aim2* that is inflammatory-regulated and not the canonical isoform defined in GENCODE annotation, nor isoforms from the RefSeq annotation.

## The novel inflammatory promoter of *Aim2* is regulated by IRF3 and P65

To gain insights into potential regulatory mechanisms controlling the expression of the 152 significant AFE events, we assessed changes in chromatin accessibility during inflammatory activation in BMDMs. Analysis of ATAC-Seq (*Tong et al., 2016*) revealed differential peaks at the promoter regions for 25% of genes with significant AFE events, suggesting that chromatin remodeling is one mechanism driving the expression of the AFE events (*Figure 3A*). This regulation is not what controls isoform usage for *Aim2*. The annotated promoter is accessible in all cells while the novel *Aim2* promoter is specific to myeloid progenitors and monocytes (*Lara-Astiaso et al., 2014*; *Tong et al., 2016*; *Figure 3—figure supplement 1A*). In addition, the accessibility of both the annotated and unannotated promoters remains open despite the cell's inflammatory status (*Figure 3—figure supplement 1B*). Therefore, the expression of the new isoform is not due to chromatin remodeling of either promoter region (*Figure 3—figure supplement 1C*).

Another potential mode of regulation that can drive AFE usage is TF binding. We next analyzed ChIP-seq data of two major TFs that drive inflammation downstream of LPS, NF-κB (p65) and interferon response factor 3 (IRF3) (*Tong et al., 2016*). We found that p65 and IRF3 specifically account for another 25% of the AFE events, including the novel *Aim2* isoform, which we confirmed using multiple ChIP-seq data sets (*Figure 3A*, *Figure 3—figure supplement 2*).

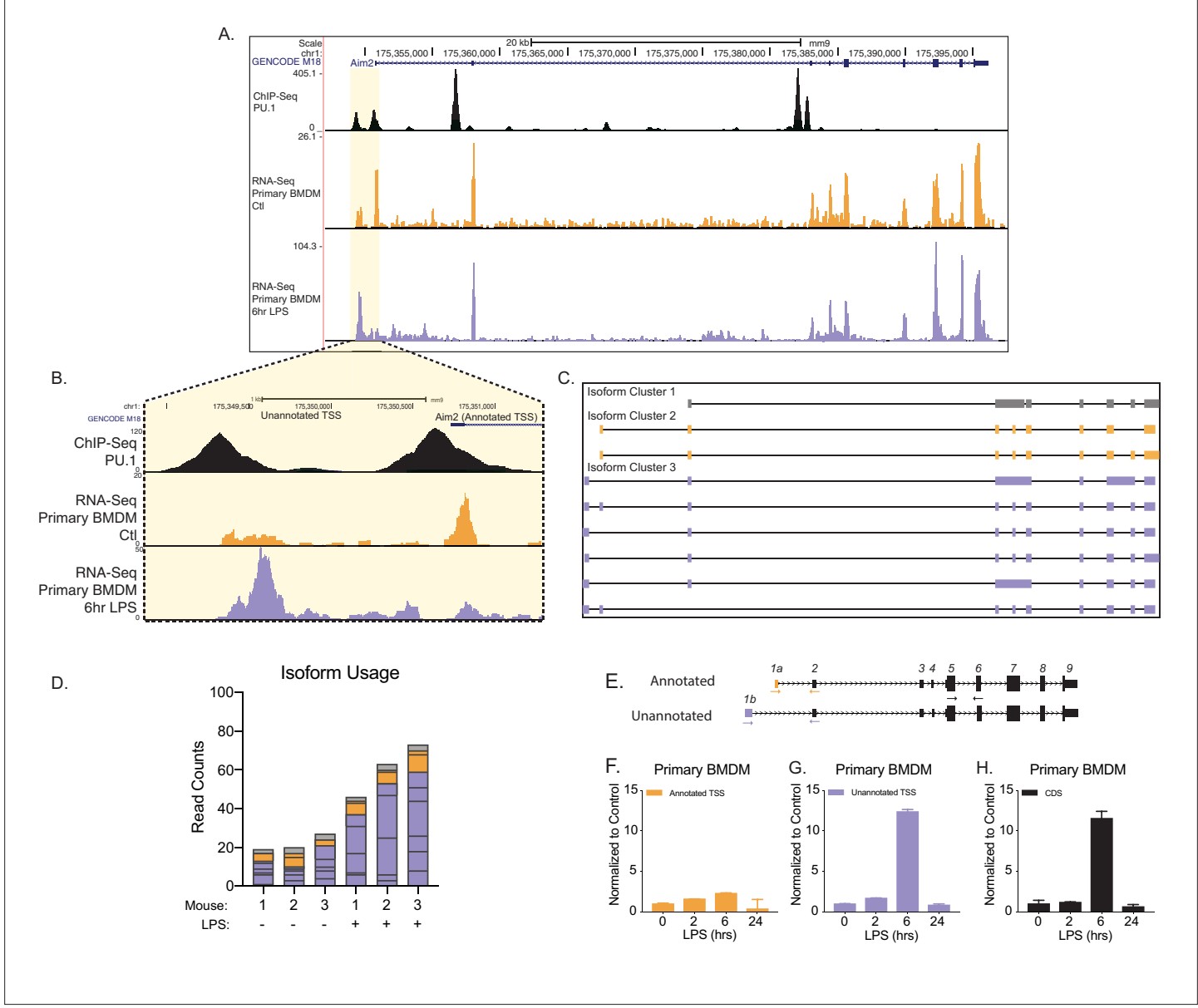

**Figure 2.** Identification of an unannotated promoter in *Aim2*. (**A, B**) The top track, in black, represents ChIP-seq data for a macrophage-specific transcription factor, PU.1. Peaks represent possible promoter regions; two distinct peaks of equal height are present at the annotated transcriptional start site (TSS) for *Aim2* and about 1 kb upstream of the transcriptional start site. The middle track, in orange, represents basal transcription in bone marrow-derived macrophages (BMDMs), while the bottom track, in purple, represents active transcription in BMDMs 6 hr lipopolysaccharide (LPS) post treatment. (**C**) *Aim2* transcript isoforms identified in BMDMs by native RNA long-read sequencing through FLAIR analysis. Transcripts are categorized by promoter, denoted by gray, orange, or purple. (**D**) The bar chart represents data from long-read sequencing showing the abundance of each transcript isoform from BMDMs ±6 hr LPS. (**E–H**) qRT-PCR was performed in biological triplicate, on primary BMDM RNA extracts that had been stimulated with LPS for indicated time points.

Further bioinformatic analysis of promoters associated with all AFEs, using HOMER TF motif enrichment, shows that there are 304 potential TFs that bind these promoters. By gene ontology analysis, we see that the majority of the TFs are associated with metabolism, as well as the immune system (*Figure 3B*; *Supplementary file 10*). When specifically assessing the two promoter regions driving the annotated and unannotated isoforms of *Aim2*, we identified 106 individual TF motifs within the annotated promoter and 121 motifs within the unannotated promoter (*Supplementary file 11*). Of these predicted motifs, there were 44 motifs unique to the unannotated promoter (*Figure 3C*). Gene ontology (*Huang et al., 2007*) analysis of TFs specific to the annotated and unannotated promoter

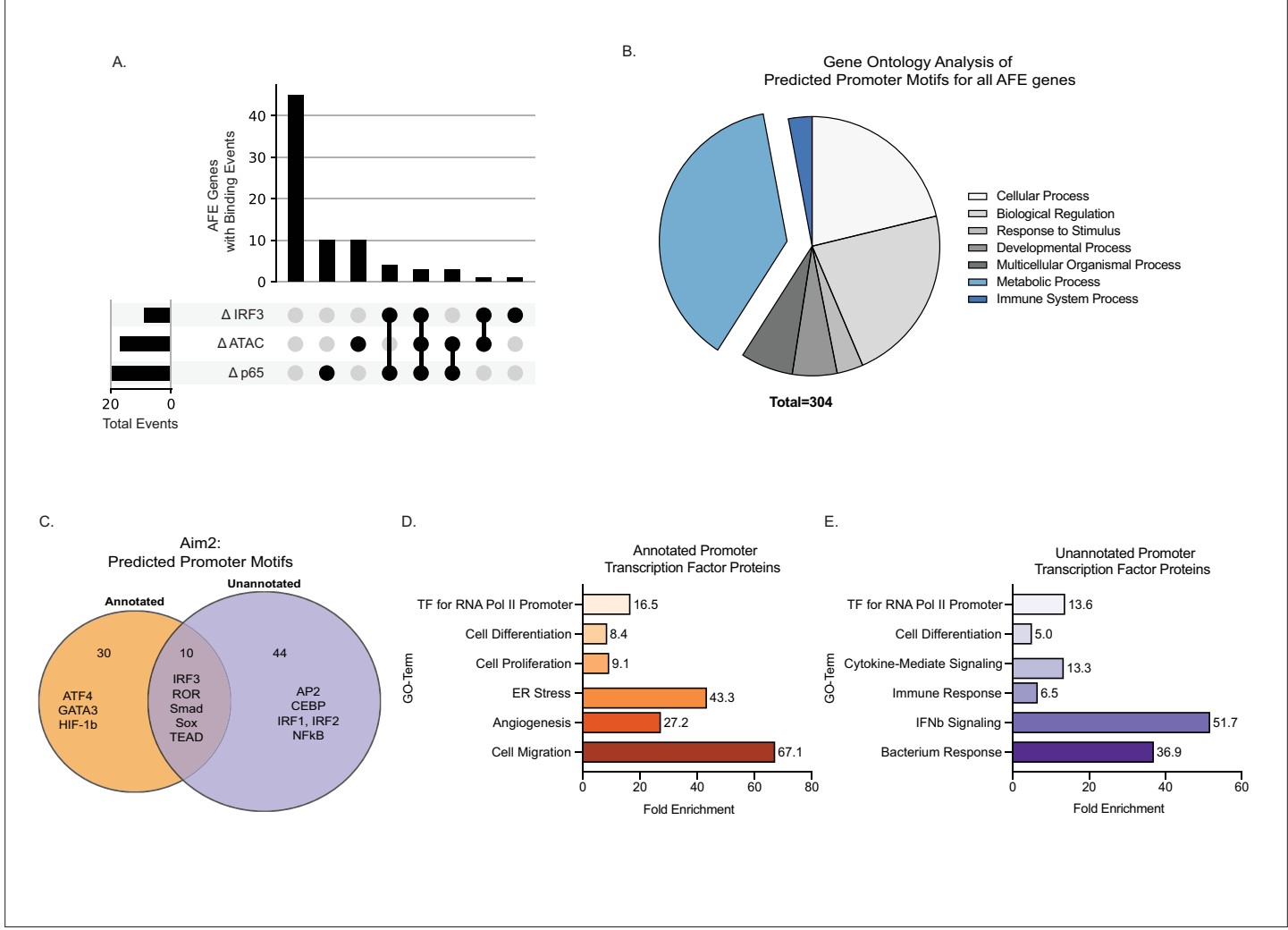

**Figure 3.** Novel inflammatory promoter of *Aim2* is regulated by IRF3 and p65. (**A**) UpSet plot showing the number of alternative first exon (AFE) events out of 77 total that have differential transcription factor (TF) binding and differential chromatin accessibility and all combinations of these sets. (**B**) Pie chart representing the gene ontology of TF motifs identified through analysis motif enrichment (AME) tool of all AFE promoter sequences. (**C**) Venn diagram of all motifs defined using HOMER analysis within the annotated and unannotated promoter regions. (**D, E**) DAVID analysis examining the gene ontology of TFs at the annotated and unannotated promoters of *Aim2*.

The online version of this article includes the following figure supplement(s) for figure 3:

**Figure supplement 1.** Chromatin remodeling not a mechanism driving novel *Aim2* isoform.

**Figure supplement 2.** Genome browser images of ChIP-seq and ATAC-seq for both the promoters of CTL and lipopolysaccharide (LPS) specific alternative first exon isoforms.

regions of *Aim2* confirms that the unannotated promoter is driven by inflammatory-specific TFs including NF-κB and IRFs (***Figure 3—figure supplement 2L***). Use of ATAC-seq and ChIP-seq for specific TFs has enabled us to determine the regulatory pathways driving 50% of the AFEs in our data. For the remaining 50%, there could be additional TFs, RNA-binding proteins (***Lynch, 2004***), or differential RNA stability driving their expression. Further work will be required to fully understand the complex regulation of all AFE events.

## Unannotated 5′UTR of *Aim2* negatively regulates translation through a single iron-responsive element

The novel inflammatory isoform identified here for *Aim2* acquired a longer 5′UTR compared to the canonical isoform; however, globally there are no differences in AFE size between conditions

(*Figure 4A*, *Figure 4—figure supplement 1A,B*). Previous studies have shown that longer 5′UTRs can affect the translation of a gene (*Kramer et al., 2013*; *Senanayake and Brian, 1999*). Using a GFP reporter system, the translational efficiency of the unannotated *Aim2* 5′UTR (767 bp) was compared to the annotated 5′UTR (489 bp) (*Figure 4B*). The unannotated 5′UTR showed significantly lower mean GFP fluorescence units, suggestive of lower translational efficiency, as assessed by flow cytometry 72 hr post-transient transfection in 293T cells (*Figure 4C,D*), while equal mCherry fluorescence was observed for all co-transfected control constructs (*Figure 4E*). To explore the mechanism of how the unannotated 5′UTR results in decreased translational efficiency, we used RegRNA2.0 to predict RNA regulatory motifs in the 5′UTRs (*Chang et al., 2013*). We identified a single IRE within the unannotated 5′UTR, while Musashi binding element (MBE) motifs were identified in both 5′UTRs (*Figure 4F*; *Supplementary file 14*). Globally, we find that there are more predicted motifs in inflammatory-regulated first exons (inclusion exons) in comparison to basal exons (excluded exons), but the IRE motif is found only in *Aim2* (*Figure 4—figure supplement 1C*). The finding that there are structured motifs in the unannotated 5′UTR of *Aim2* is also supported by the RNAfold Vienna package (*Gruber et al., 2008*), which predicts the hairpin structure of the IRE in the alternative 5′UTR (*Figure 4—figure supplement 2*). Since the IRE motif is unique to the unannotated 5′UTR of *Aim2,* we hypothesized that this motif is critical in regulating translational efficiency.

When cells are at homeostasis, iron binding proteins (IRP1/2) bind to IREs located within the 5′UTR (e.g., ferritin) and can block translation, while IREs in the 3′UTR (e.g., transferrin receptor) can promote translation (*Rouault, 2006*; *Wang et al., 2004*). However, iron repletion results in the inactivation of IRP1/2 (*Outten, 2017*; *Figure 4G*). To experimentally test if the IRE motif within the unannotated 5′UTR acts as a translational repressor, we removed the element using site-directed mutagenesis, which led to an increase in GFP expression by ~20% compared to the annotated 5′UTR (*Figure 4H,I*). Next, we exogenously added 100 μM ferric ammonium citrate (FAC) to overload the cells with iron and determined if this can rescue the observed decrease in translational efficiency from our unannotated 5′UTR. Upon FAC administration, the relative GFP expression of the unannotated 5′UTR plasmid increased by ~50%, while mCherry control was unchanged, suggesting that the translational efficiency of the unannotated 5′UTR can be rescued with iron supplementation (*Figure 4J,K*). From these results, we conclude that the predicted IRE motif within the unannotated 5′UTR of *Aim2* functions as an IRE to control translation.

To test if the IRE motif in the unannotated 5′UTR of *Aim2* acts as a translational regulator endogenously, we performed polysome profiling followed by RT-qPCR on primary BMDMs in the presence and absence of LPS for 18 hr to determine the translational competency of the isoforms of *Aim2* (*Figure 4L,M*). As a negative control, we examined *Neat1* (*Nakagawa et al., 2014*), a long noncoding RNA (lncRNA) that is not detected in polysomes, nor translated, and this is not dependent on LPS (*Figure 4N*). The relative distribution of our positive control gene, *Gapdh*, which encodes a highly expressed housekeeping protein, is enriched in the high polysome fraction, with or without LPS, as expected (*Figure 4O*). Using isoform-specific primer sets (*Figure 2E*) for the annotated and unannotated *Aim2* isoforms, we find that the annotated isoform is enriched in the high polysome fraction with and without LPS treatment (*Figure 4P*), while the novel isoform is enriched in the low polysome fraction with and without inflammatory stimulation (*Figure 4Q*). Additionally, from our RNA-seq data we find that IRP1 and IRP2 are not transcriptionally regulated by inflammation, but based on Weiss et al. inflammation can affect the binding affinity of IRPs in macrophages (*Weiss et al., 1997*; *Figure 4—figure supplement 3A–D*). These data show that the unannotated *Aim2* isoform has a lower translational efficiency compared to the canonical form.

To further validate the effect the post-transcriptional mechanism of the unannotated *Aim2* isoform, we performed time-course LPS stimulations. In primary BMDMs, a 72 hr, time-course stimulation with and without the treatment of iron (FAC) protein lysates was generated and Aim2 expression was measured by western blot (*Figure 4R-SFigure 4—source data 1*). Aim2 is expressed basally and significantly decreases upon LPS treatment at the 48 hr time point, most likely as a control mechanism to return the pathway to homeostasis and limit the inflammatory stimulation. When FAC is added to cells, Aim2 expression does not decrease at the 48 hr time point suggesting that it is indeed the IRE that is driving this decrease in Aim2 observed in the wild type cells (*Figure 4R-SFigure 4—source data 1*). Finally, we utilized the CRISPR/Cas9 technology to remove the Aim2 IRE endogenously (*Figure 4—figure supplement 4A*). In these immortalized BMDMs, RT-qPCR was performed

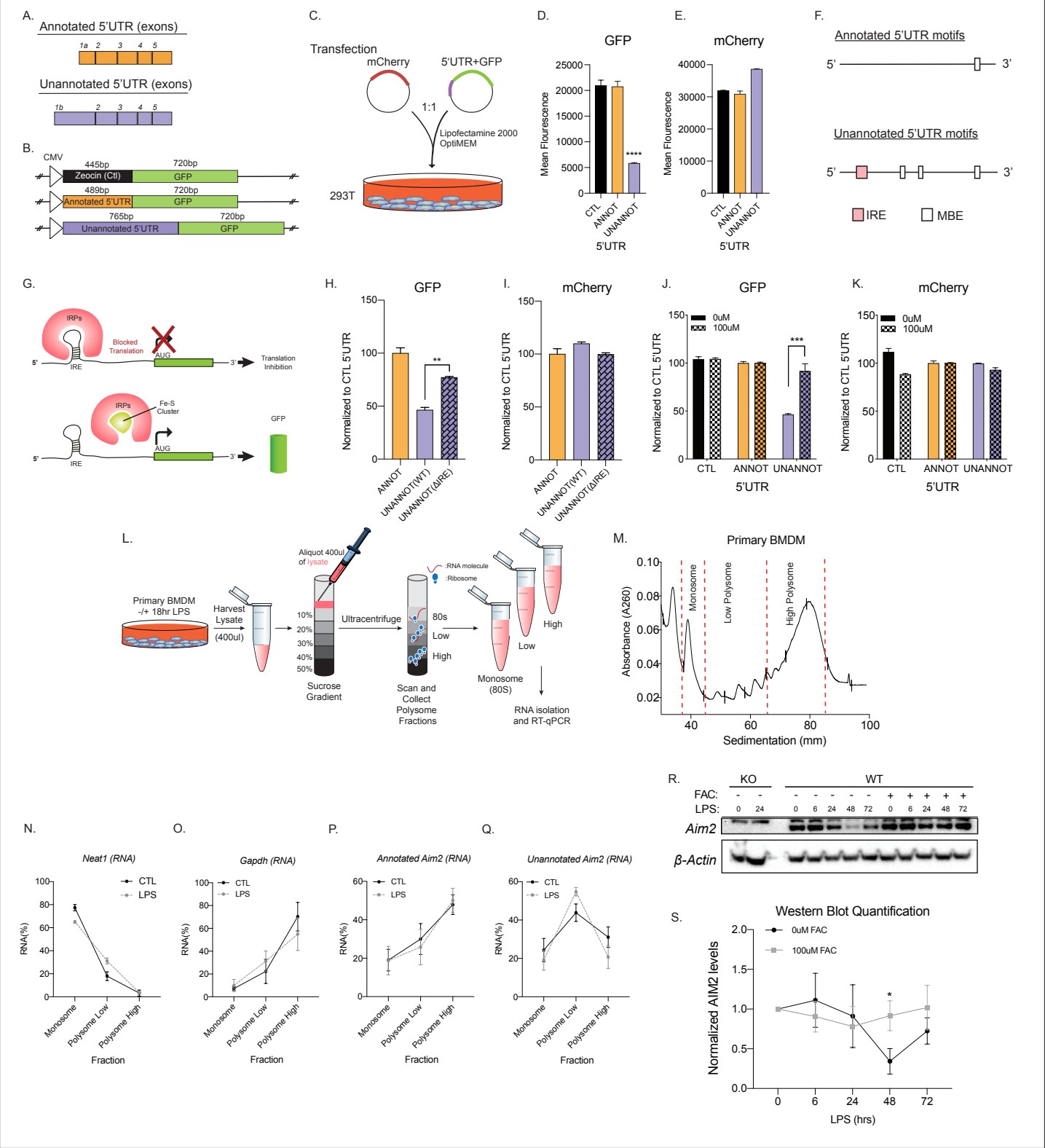

**Figure 4.** Unannotated 5' untranslated region (UTR) of *Aim2* negatively regulates translation through a single iron-responsive element (IRE). (**A**) Schematic of annotated and unannotated 5'UTR of most prevalent *Aim2* isoforms in mouse macrophages. (**B**) Diagram of cloning strategy of *Aim2*'s 5'UTR in GFP plasmid. (**C**) Transfection strategy of 5'UTR-GFP plasmids co-transfected with an mCherry control plasmid at a 1–1 ratio in 293T cells. (**D, E**) Flow cytometry of 293T cells 72 hr post-transfection with control annotated and unannotated 5'UTR of *Aim2* to measure GFP and mCherry (Ctl) protein fluorescence. (**F**) Using RegRNA2.0, a single IRE was found in the alternative 5'UTR, in addition to multiple Musashi binding elements (MBEs).

*Figure 4 continued on next page*

*Figure 4 continued*

(**G**) Diagram of how an IRE functions in the cytoplasm of a cell within a 5'UTR. With low or normal levels of iron, iron binding proteins (IRP1 or IRP2) bind to IRE and block translation. During high levels of iron, within a cell, IRP1 is sequestered by iron-sulfur (Fe-S) clusters and IRP2 is degraded, therefore allowing translation of the protein. (**H, I**) Flow cytometry of 293T cells 72 hr post-transfection of mCherry (Ctl), along with an annotated 5'UTR-GFP plasmid, unannotated 5'UTR-GFP plasmid, or a GFP plasmid containing the unannotated 5'UTR without the defined IRE. (**J, K**) Flow cytometry of 293T cells ± 100 μM ferric ammonium citrate (FAC) 72 hr post-transfection of mCherry (Ctl), along with an annotated 5'UTR-GFP plasmid or unannotated 5'UTR-GFP plasmid. (**L**) Overview of the polysome profiling protocol to analyze translation activity. (**M**) Cytoplasmic lysates from ±lipopolysaccharide (LPS)-treated cells were fractionated through sucrose gradients. Global RNA polysome profiles generated by the density gradient fractionation system are shown. A representative plot from stimulated primary bone marrow-derived macrophage (BMDM)-fractionated samples is shown. The experiment was performed three times. (**N–Q**) The relative distribution of *Gapdh* mRNA, encoding a housekeeping protein, *Neat1*, long non-coding RNA (lncRNA), annotated and unannotated *Aim2* mRNA, was measured by RT-qPCR analysis of RNA. Each of the gradient fractions are calculated as relative enrichment when compared to unfractionated input mRNA; standard deviation represents technical triplicate. (**R**) Protein lysates of time-course LPS stimulation of 0 hr, 6 hr, 24 hr, 48 hr, and 72 hr without and with 100 μM of FAC (iron) added to immortalized WT BMDMs. Western blot performed on AIM2 and B-ACTIN. (**S**) Western blot quantification performed in FIJI, standard deviation represents biological triplicates, and p-value assessed using Student's t-test.

The online version of this article includes the following figure supplement(s) for figure 4:

**Figure supplement 1.** Global characterization of alternative first exon (AFE).

**Figure supplement 2.** *Aim2* iron-responsive element (IRE) hairpin structure predicted.

**Figure supplement 3.** IRP1 and IRP2 are not differentially expressed by lipopolysaccharide (LPS) in bone marrow-derived macrophages (BMDMs).

**Figure supplement 4.** CRISPR/Cas9 knock-out of iron-responsive element (IRE) in *Aim2* 5′ untranslated region (UTR) leads to an increase of protein expression in mouse macrophages during inflammation.

**Source data 1.** Supplemental WB Uncrop Primary BMDM +/- Iron.

**Figure supplement 4—source data 1.** Uncropped western blot images from WT Cas9 BMDM cell line from *Figure 4—figure supplement 4F*.

**Figure supplement 4—source data 2.** Uncropped Western Blot Images for IRE KO BMDM Cas9 Cell Line.

with an LPS time course. The annotated *Aim2* isoform shows no induction by LPS (*Figure 4—figure supplement 4C*), while the unannotated *Aim2* isoform and the CDS sequence are induced by twofold (*Figure 4—figure supplement 4D,E*). There is no significant difference in transcriptional expression at the *Aim2* locus between the WT and IRE KO cell lines (*Figure 4—figure supplement 4D,E*). Using these newly characterized WT and IRE KO clonal cell lines, we performed a 72 hr time course of LPS stimulation (*Figure 4—figure supplement 4F,G, Figure 4—figure supplement 4—source data 1*). At the 48 hr time point, there is a significant decrease in expression of Aim2 protein in WT cells, but not in the IRE KO cells (*Figure 4—figure supplement 4H*), further indicating that endogenous Aim2 is regulated through the IRE motif located in the AFE.

# Discussion

While we have come a long way in determining the transcriptomes of immune genes to better under-stand signaling pathways, very little work has focused on the role that mRNA isoforms play (*Akira et al., 2006*; *Medzhitov and Janeway, 1998*; *Pai et al., 2016*). Over 95% of genes have more than one mRNA transcript due to alternative splicing, but the regulatory importance of these splicing events is not fully understood (*Carninci et al., 2006*; *Tian and Manley, 2017*; *Wang et al., 2008*). On a gene-by-gene approach, alternative splicing has been shown to play a role in health and disease by shaping the proteome (*Nishiyama et al., 2000*; *Yang et al., 2011*; *You et al., 2016*). Globally, a number of labs have tackled the prevalence of alternative splicing in vitro and in vivo, showing that alternative splicing can affect both the nature and duration of inflammation (*Beyer et al., 2012*; *Bhatt et al., 2012*; *Janssen et al., 2020*). To date, no one has examined the conservation of these mechanisms using primary cells or utilized long-read sequencing to build the transcriptome de novo to obtain a complete understanding of the extent of alternatively expressed isoforms generated following an immune response.

In our study, we demonstrate a conservation of splicing, specifically AFE events in both human and murine macrophages. We found that there are 11 genes that have AFE in both human and mouse macrophages (*Figure 1—figure supplement 5A*). Most studies to date have focused on isoform changes linked to genes that are differentially expressed following inflammation, and interestingly these 11 genes would have been previously overlooked because many of them are not differentially

expressed, emphasizing the importance in studying isoform expression in all conditions (*Figure 1E,F*). Of the 11 conserved genes, 7 AFE isoforms have been previously studied in some context including *Rps6ka1* (*Qin et al., 2018*), *Ncoa7* (*Singer et al., 2008*; *Yu et al., 2015*), *Rcan1* (*Pang et al., 2018*), *Wars* (*Liu et al., 2004*; *Miyanokoshi et al., 2013*), *Arap1* (*Kulzer et al., 2014*), *Tsc22d1* (PMID:21448135), and *Sgk1* (*Arteaga et al., 2007*; *Kobayashi et al., 1999*; *Lang et al., 2006*). While this validates our technique, it is important to note that none of these genes had been connected to inflammation or formally shown to be conserved mechanisms of regulation, besides *Wars* (*Liu et al., 2004*; *Miyanokoshi et al., 2013*). This also highlights our method's ability to accurately identify inflammatory regulated RNA isoforms, in addition to the uncharacterized AFE events of *Ampd3*, *Snx10*, *Shisa5*, and *Tspan4*. Furthermore, *Snx10* is studied outside the context of inflammation, has been implicated in chronic inflammatory disease, and our study may suggest new insights into how alternative splicing could be regulating these genes (*You et al., 2016*). We further validated *Ncoa7*, *Rcan1*, and *Ampd3* in human and murine macrophages using RT-PCR (*Figure 1—figure supplement 6*).

To overcome the current limitations of any transcriptome build, we used direct RNA nanopore technology on primary murine macrophages to build our own transcriptome de novo with the aim of identifying novel full-length transcriptional isoforms that are expressed with and without inflammation (*Tang et al., 2020*; *Workman et al., 2019*). The limitation of direct RNA-sequencing is read depth, and our average read depth was about 1 million reads per treatment. In a previous study of direct RNA-sequencing, we found that increased sequencing depth will result in additional isoforms detected (*Workman et al., 2019*). Therefore, we expect to see additional unannotated AFE and other AS events with increased sequencing depth. However, even with 1 million reads, we were able to build a transcriptome that led to the identification of 154 novel AS events. Following this, we identified hundreds of novel isoforms resulting in 95 novel AFE events, supported by CAGE-seq (*Figure 1*, *Figure 1—figure supplements 7 and 8*), including an unannotated mRNA isoform of the well-studied gene protein absent in melanoma 2 (*Aim2*).

*Aim2* is characterized as an interferon-inducible gene (*Johnstone and Trapani, 1999*) (PMID:10454530), functioning as a cytoplasmic dsDNA sensor leading to the formation of an inflammasome and eventual cleavage and release of pro-inflammatory cytokines of IL1β and IL18 (*Bürckstümmer et al., 2009*; *Fernandes-Alnemri et al., 2009*; *Hornung et al., 2009*). Our study highlights that it is an alternative mRNA isoform of *Aim2* that is inducible, and that this upregulated transcript is translated less efficiently compared to the canonical isoform. This novel finding goes against the existing assumption that induced gene expression results in induced protein expression (*Figures 1I, 2 and 4*, *Figure 4—figure supplement 4*). FLAIR-identified transcripts (*Workman et al., 2019*) show three clear AFEs for *Aim2,* and only one of those first exons is inflammatory inducible (*Figure 2C,D*). RT-qPCR further confirms that the annotated TSS of *Aim2* is not inducible, while the unannotated AFE of *Aim2* is LPS inducible (*Figure 2E–H*). This result highlights the need for cell-type and treatment-specific transcriptome annotations if one is to have a complete understanding of the transcriptome and proteome of a given cell.

We further investigated what drove the expression of this novel *Aim2* isoform, as well as what drove the expression of all the AFE genes. Using ATAC-seq (*Lara-Astiaso et al., 2014*; *Tong et al., 2016*) and ChIP-seq (*Tong et al., 2016*) data sets, we were able to determine that 50% of the AFE events were driven partially by chromatin accessibility and inflammatory-specific TFs (*Figure 3A*). Using HOMER, a TF motif enrichment tool, we identified that the majority of TFs that regulate the expression of the AFE genes are involved in metabolism pathways (*Figure 3B*). Additionally, another mechanism that can drive AFE usage is the splicing of internal exons that activate proximal upstream promoters (*Fiszbein et al., 2019*). Further analysis incorporating more TFs and coupling multiple splicing events will be necessary to determine the definitive regulatory mechanism of all AFEs. Interestingly, the annotated *Aim2* promoter accessibility is constitutively open across all hematopoietic cells, while the unannotated *Aim2* promoter is only accessible in myeloid progenitors or terminally differentiated cells (*Figure 3—figure supplement 1A*), meaning that the novel *Aim2* isoform may be specific to myeloid cells. Furthermore, *Aim2* annotated and unannotated promoter usage is not driven by chromatin accessibility (*Figure 3—figure supplement 1B,C*) but is driven by the activation of inflammatory-specific TFs (*Figure 3C–E*, *Figure 3—figure supplement 2L*).

There is no difference in the open reading frame of the novel isoform of Aim2 when compared to the annotated transcript using NCBI ORFfinder; therefore, we turned our attention to a possible

regulatory mechanism within the 5′UTR (*Mignone et al., 2002*). Broadly, UTRs play crucial roles in the post-transcriptional regulation of gene expression, including alteration of the mRNA translational efficiency (*van der Velden and Thomas, 1999*), subcellular localization (*Jansen, 2001*), and stability (*Bashirullah et al., 2001*). Post-transcriptional regulatory mechanisms of Aim2 have not been previously studied. Using RegRNA2.0 (*Chang et al., 2013*) globally, we identify that LPS-activated AFEs contain more regulatory elements than basal first exons (*Figure 4—figure supplement 1*). More importantly, we identified an IRE, a regulatory motif unique to only the novel 5′UTR of *Aim2* (*Figure 4F*). Utilizing a GFP reporter plasmid, we were able to determine that the IRE motif was functional, by recapitulating the same experiments used on the protein Ferritin, the first functional IRE motif ever studied (*Leedman et al., 1996*). Then, we showed the inflammatory-specific mechanism regulating Aim2 protein expression by performing a western blot on primary BMDMs with and without FAC during a 72 hr LPS time-course experiment. Aim2 protein is basally expressed, and while the transcript is inducible, specifically the novel isoform, we do not observe an increase in expression of Aim2 protein by western blot. In fact, we find that Aim2 protein decreases following inflammation, and this can be reversed by iron supplementation. Finally, we confirmed the IRE mechanism by using CRISPR/Cas9 to generate an IRE KO clonal cell line. Once our cell line was characterized (*Figure 4—figure supplement 4A–E*), we performed an LPS time-course stimulation and measured Aim2 protein expression. At the 48 hr time point, we see a significant rescue of Aim2 protein expression in the IRE KO cell line (*Figure 4—figure supplement 4F–H*). This could be a critical regulatory step that has evolved to ensure the Aim2 pathway is switched off following its formation and activation of the inflammasome.

These results demonstrate that the inflammatory-specific mRNA isoform of *Aim2* has lower translational efficiency than the canonical form and that protein translation can be increased by the addition of iron. *Crane et al., 2014* demonstrate that ROS can contribute to activation of Aim2 inflammasome in mouse macrophages. Our proposed mechanism of translational regulation of Aim2 is through an IRE, which is known to directly interact with IRP proteins (*Outten, 2017*; *Rouault, 2006*; *Wang et al., 2004*). Interestingly, IRP2 degradation can be driven not only through iron, but also through ROS and RNS (*Hanson and Leibold, 1999*), further supporting this novel IRE regulatory mechanism of Aim2 protein expression. Finally, Cheng et al. have shown that AIM2 is regulated by oxidative stress and show that overactivation of AIM2 inflammasome can contribute to pancreatic tumorigenesis, all within the environment of mitochondrial iron overload (*Li et al., 2018*). This newly identified isoform, with an IRE-specific translational mechanism, provides mechanistic understanding to these recent studies of Aim2 (*Crane et al., 2014*; *Li et al., 2018*). These findings could have significance for better understanding the mechanisms driving pathology in inflammatory disease such as systemic lupus erythematosus (SLE) (*Corbett, 2018*). AIM2 expression levels have been correlated with severity of inflammation in SLE patients (*Zhang et al., 2013*), and it is well known that iron is dysregulated in this disease (*Vanarsa et al., 2012*). It is possible that AIM2 levels remain high in SLE patients due to dysregulated iron; therefore, homeostasis in macrophages cannot be maintained.

In summary, signaling within macrophages enables us to fight infection but also can contribute to pathological inflammation associated with a wide variety of diseases. While there are multiple regulatory checkpoints in place to control inflammation, we propose that alternative splicing and translational regulation play critical roles in maintaining this type of control. A better understanding of the molecular mechanisms that control inflammatory-regulated genes, including *Aim2*, could provide new targets for therapeutic intervention of autoimmune and inflammatory diseases.

## Materials and methods
### Maintenance of mice
UCSC and the Institutional Animal Care and Use Committee maintained mice under specific pathogen-free conditions in the animal facilities of University of California Santa Cruz (UCSC) in accordance with the guidelines.

## Human PBMC-derived macrophage differentiation and in vitro stimulation

Human peripheral blood mononuclear cells (PBMCs) were enriched by density gradient centrifugation of peripheral blood from healthy human donors through a Ficoll-Paque PLUS (GE Healthcare) gradient. Monocytes were isolated from PBMC by negative selection using the EasySep Human Monocyte Isolation Kit (STEMCELL Technologies) according to the manufacturer's instructions. To differentiate monocytes into macrophages, recombinant human M-CSF (50 ng/mL) was used in RPMI-1640 medium with 10% fetal bovine serum (FBS), 2 mM L-glutamine, 10 mM HEPES, 1 mM sodium pyruvate, 100 U/mL penicillin, and 100 µg/mL streptomycin. The culture medium that contained fresh recombinant human M-CSF was replaced every 2 days.

## Cell culture, mouse macrophage differentiation and stimulation

Cells were cultured in DMEM with 10% FBS supplemented with penicillin/streptomycin or ciprofloxacin. Primary BMDMs were generated by cultivating erythrocyte-depleted bone marrow cells in the presence of 30% L929 supernatant, and the cells were used for experiments 6–9 days after differentiation. J2Cre virus (*Blasi et al., 1985*) was used on day 3/4 after isolation of bone marrow cells to establish transformed BMDM cell lines. BMDMs were cultivated in the presence of J2Cre virus for 48 hr, and L929 was then gradually tapered off over 6–10 weeks depending on the growth pattern of transformed cells.

## In vitro stimulation of macrophages

Bone-derived macrophage cells were primed with 100 µM of FAC for 24 hr prior to TLR stimulation. BMDM cells were stimulated with TLR ligands for the indicated time points using the following concentrations: LPS 200 ng/mL (TLR4). For RNA and protein isolation, $1–2 \times 10^6$ cells were seeded in 12-well format.

## RT-PCR validation

RT-PCR validation was completed using three biological replicates. KAPA HiFi HotStart ReadyMix PCR Kit (Kapa Biosystems) and the manufacturer's suggested cycling protocol were used to complete the PCR reaction with the following primers:

> *Mse_Denr_F1:* ATCGCGATAAAGGCTCATTG
> *Mse_Denr_F2:* GCTACCTGTCCTTTTCCCCA
> *Mse_Denr_R:* AACTTGGCACTGTTCTTCGT
> *Mse_Arhgef7_F1:* TGTTGTTCTGGGGTTTGTGA
> *Mse_Arhgef7_F2:* CTGTGTGTTGCAGGTCTACC
> *Mse_Arhgef7_R:* GTGTCACCAAGGAGCTGAGG
> *Mse_Ncoa7_F1:* GTGGTGGAGAAGGAAGAGCT
> *Mse_Ncoa7_F2:* TTCTATTGTGCCAGGCCTGA
> *Mse_Ncoa7_R:* GCATGTTTTCCAGGAGTGCA
> *Mse_Ampd3_F1:* CCCTACTGTAGATGAATCCCCTTA
> *Mse_Ampd3_F2:* GCTGAGCTTTGTGTCTGTGT
> *Mse_Ampd3_R:* GGGGACAGTAAACAGGGACA
> *Mse_Rcan1_F1:* ACTGGAGCTTCATCGACTGC
> *Mse_Rcan1_F2:* GACTGAGAGAGCGAGTCGTT
> *Mse_Rcan1_R:* CATCGGCTGCAGATAAGGGG
> *Hu_NCOA7_F1:* TGTTCAGTGGTCTCCCGATGTCTATGG
> *Hu_NCOA7_R:* GGGCCGTAGGACAGGCAGCA
> *Hu_NCOA7_R2:* AGCGTGGCTACAAGTAACTGTGGTGT
> *Hu_AMPD3_F1:* TATGCAAAACAGAGACCTCC
> *Hu_AMPD3_R:* CACTTCAGAGATGTTCAGCT
> *Hu_AMPD3_F2:* CCTGCTTGGTTTTAGAGGAT
> *Hu_RCAN1_F1:* GACTGGAGCTTCATTGACTG
> *Hu_RCAN1_R:* ATTCTGACTCGTTTGAAGCT
> *Hu_RCAN1_F2:* TAGCGCTTTCACTGTAAGAA

Band intensities were measured for each band in each condition and sample using FIJI (*Schindelin et al., 2012*). The relative abundance of each isoform was calculated using the equation to calculate

percent spliced in (PSI) (PSI = inclusion/(inclusion + exclusion)) in each condition and sample to vali-
date the computationally derived delta PSI values. A gel extraction was completed for each band
using the PCR clean-up Gel extraction kit (Machery-nagel). The PCR product was confirmed using
Sanger Sequencing.

## RNA isolation, RT-qPCR

Total cellular RNA from BMDM cell lines or tissues was isolated using the Direct-zol RNA MiniPrep Kit
(Zymo Research) according to the manufacturer's instructions. RNA was quantified and controlled for
purity with a nanodrop spectrometer (Thermo Fisher). For RT-qPCR, 500–1000 ng were reversely tran-
scribed (iScript Reverse Transcription Supermix, Biorad) followed by RT-PCR (iQ SYBRgreen Supermix,
Biorad) using the cycling conditions as follows: 50°C for 2 min, 95°C for 2 min followed by 40 cycles of
95°C for 15 s, 60°C for 30 s, and 72°C for 45 s. The melting curve was graphically analyzed to control
for nonspecific amplification reactions. Quantitative RT-PCR analysis was performed with the following
primers:

*Mse_Aim2_F_Annotated*: CCGCCATGCTTCCTTAACTA
*Mse_Aim2_F_Unannotated*: AGGCGGATGGTTTGAACTCT
*Mse_Aim2_R_Exon2*: TTGAAGCAACTTCCATCTGC
*Mse_Aim2_CDS_F*: AGTACCGGGAAATGCTGTTG
*Mse_Aim2_CDS_R*: GAGTGTGCTCCTGGCAATCT
*Mse_Gapdh_F*: CCAATGTGTCCGTCGTGGATC
*Mse_Gapdh_R*: GTTGAAGTCGCAGGAGACAAC
*Mse_Neat1_F*: TTGGGACAGTGGACGTGTGG
*Mse_Neat1_R*: TCAAGTGCCAGCAGACAGCA

## Cloning strategy for 5'UTR GFP plasmid

The GFP reporter plasmid was CMV-Zeo-t2A-GFP. Zeocin is flanked by NheI and AgeI. The sequence of
the annotated and unannotated 5'UTR was used as defined by the UCSC RefSeq and our sequencing
results to be. Using KAPA HiFi HotStart ReadyMix PCR Kit (Kapa Biosystems), the two 5'UTRs were
amplified from cDNA.

| 5'UTR | cDNA sequence |
| --- | --- |
| Annotated | TTCCTGTCCTGTCTGCCGCCA TGCTTCCTTAACTAGCTGCTA GGTTTTTTCCTTGTCGTGATG AAATCCACCCTCATGGACCT GTAAGTAAAATGTAGACTTG CATAGAGTGCTGTAATCTTAC GGCCGAGGTTTCTTTTCAGG CTGATCCTGGGACTGTGAG |
| Unannotated | TATATCTAAAATACCTCTGGTT GAGACCTCACAGCTGGAGGAG AAACTCTGCTGAGGCTTGTAA AAAGGAAACTGAAAACTAGCA TTTGCTTGGGCAGAGCCTTAATAT ATAATTATTTTGCCCCAGCATCAGG GTTTAGGACTCAGCTATAGGGCCA GGACTAGCCAAGCTTCAAAGTGA AAGAAGATAGTTGAGAGTACTTTC TGCTTTCTGTCTCCCAAGACCTGAT TTTCATGATTTTCATGTCCTACTACT CATAGTGAAAATCTTTGTGAGGCGG ATGGTTTGAACTCTCAGGACATACA CCAGTCCCTGAGTTGAGAACTAAGG CTGCTTTGGAGAGAAGAAAATCCCC TGAGGTAAGTAGACTTGCATAGAGTG CTGTAATCTTACGGCCGAGGTTTCTTT TCAGGCTGATCCTGGGACTGTGAG |

## Primers

*cDNA_F_Annotated:* TTCCTGTCCTGTCTGCCG

*cDNA_F_Unannotated:* TATATCTAAAATACCTCTGGTTGAGACCTC
*cDNA_R_5'UTR:* CTCACAGTCCCAGGATCAGC

The 5′UTRs were then PCR amplified with primers containing restriction enzyme sites for AgeI and NheI.

*NheI_F_Annotated:* GGT**GCTAGC**TTCCTGTCCTGTCTGCC
*NheI_F_Unannotated:* GGTGGT**GCTAGC**TATATCTAAAATACCTCTGGTTGAGA
*AgeI_R_5'UTR:* GGT**ACCGGT**CTCACAGTCCCAGGATCAGC

PCR products, as well as the GFP plasmid, were then processed using AgeI and NheI restriction enzymes overnight. These samples were run on a 1% agarose gel. A gel extraction was completed for each band using the PCR clean-up Gel extraction kit (Machery-nagel). The PCR product was confirmed using Sanger Sequencing.

## Site-directed mutagenesis

Set up PCR reaction with 1 µL 279 plasmid, with unannotated Aim2 5′UTR, 5 µL 10 × Phu buffer, 1 µL F primer (0.1 µg/µL) [Remove_IRE_F - CCCTGATGCTGGGGCAAAATAATTATAAATGCTAGTTTTC AGTTTC], 1 µL R primer (0.1 µg/µL) [Remove_IRE_R - GAAACTGAAAACTAGCATTTATAATTATTTTG CCCCAGCATCAGGG], 1 µL dNTP (10 nM), 1 µL Phu polymerase, and 40 µL dH$_2$O. PCR program: 95°C 1 min, 18 cycles of 95°C 30 s, 55°C 1 min, 72°C 1 min, then end PCR with 72°C 1 min and 4°C hold. Add 0.5 µL of Dpn1 (NEB) to 25 µL PCR reaction. Incubate at 37°C for 1 hr to digest parental DNA. Transform digested and undigested plasmid into DH5α competent cells. Pick ~10–15 colonies and start overnight cultures. Colony PCR plasmids using Dpn1_Colony_PCR_F - TTGGCTAGTCCTG-GCCCTAT and Dpn1_Colony_PCR_R - GCTGGTTTAGTGAACCGTCAG to check for 20 bp deletion on a 3% agarose gel. Grow up colonies that have deletion, miniprep plasmids and send to Sequetech for sequence verification.

## Transfection of 5′UTR GFP and mCherry plasmid

A 1:1 ratio of the GFP vector containing the mature sequence of Aim2 5′UTR (annotated or unannotated) or zeocin and a plasmid containing mCherry were transfected into 293T cells for 48–72 hr. A 6-well plate of HEK293Ts was plated the night before with a concentration of 2 × 10$^5$. HEK293Ts cells were primed with 100 µM of FAC for 24 hr prior to transfection. Transfection was performed on HEK293Ts (±100 µM FAC) using lipofectamine 2000, serum-free OPTI-MEM media was used as a transfection reagent according to the manufacturer's instructions, and a (1:1) concentration of the 5′UTR GFP reporter plasmid and the mCherry control plasmid. HEK293Ts were visualized via flow cytometry 48–72 hr post transfection.

## Polysome profiling

Prior to lysis, cells were treated with cycloheximide (100 mg/mL), 10 min at 37°C, 5% CO$_2$. Cells were washed three times with ice-cold PBS and lysed in ice cold buffer A (0.5% NP40, 20 mM Tris HCl pH 7.5, 100 mM KCl, and 10 mM MgCl$_2$). Lysates were passed three times through a 23G needle and incubated on ice for 7 min. Extracts were then centrifuged at 10K rpm for 7 min at 4°C. The supernatant was collected as crude cytosolic extract. Cytosolic extracts were overlaid on 10–50% sucrose gradients prepared in 20 mM Tris HCl pH 7.5, 100 mM KCl, and 10 mM MgCl$_2$ buffer (prepared using the Gradient Station, Biocomp Instruments). Gradients were then ultracentrifuged at 40K rpm for 1 hr 20 min at 4°C using an SW41 in a Beckman ultracentrifuge. Individual polyribosome fractions were subsequently purified using a Gradient Station (Biocomp Instruments) and stored in (1:3) TRI Reagent.

## IRE KO cell line generation

The gRNA construct was constructed from a pSico lentiviral backbone driven by EF1a promoter expressing T2A flanked genes: puro and cherry. gRNAs were expressed from a mouse U6 promoter. Cloning of 20 nt gRNA spacer forward/reverse oligos was annealed and cloned via the AarI site.

*IRE_gRNA1_F:* TTGGACTGAAAACTAGCATTTGCT
*IRE_gRNA1_2:* AAACAGCAAATGCTAGTTTTCAGT
*IRE_gRNA2_F:* TTGGCTGAAAACTAGCATTTGCTT
*IRE_gRNA2_2:* AAACAAGCAAATGCTAGTTTTCAG

IRE_gRNA3_F: TTGGGGCAAAATAATTATATATTA
IRE_gRNA3_2: AAACTAATATATAATTATTTTGCC

3 µg of pooled gRNAs were electroporated using the Lonza Amaxa Mouse Macrophage Nucleofector kit (VPA-1009). Electroporated cells were then selected using puromycin 5 µg/mL and grown to 80% confluency. Limited dilution series were seeded in 96-well plate, let to grow for 3 weeks. Then the clonal cell lines were genotyped using: F:GCAGGAAATAACTTTTGTGGAGT and R:TGGGGAGAC AGAAAGCAGAAAG. Then this PCR product was sequenced, and KO efficiency was assessed using ICE Synthego (https://ice.synthego.com/#/). Then the hairpin structure was assessed by RNAfold (http://rna.tbi.univie.ac.at/cgi-bin/RNAWebSuite/RNAfold.cgi).

## Protein lysate and western blot

Cell lysates were prepared in RIPA buffer (50 mM Tris-HCl pH 8.0, 150 mM NaCl, 1 mM EDTA, 1 % [v/v] NP-40, 0.5% [w/v] sodium deoxycholate, 0.1% [w/v] SDS) containing protease inhibitor cocktail (Roche) and quantified by the Bicinchoninic Acid Assay (BCA) assay (Thermo Fisher). Equivalent masses (20 µg) of each sample were resolved by SDS-PAGE and transferred to a polyvinylidene difluoride (PVDF) membrane and western blotted with either Aim2 (1:1,000; Cell Signaling #63660) or horseradish peroxidase-conjugated b-actin monoclonal antibody (1:5000, Santa Cruz Biotechnology) used as a loading control. HRP-conjugated goat anti-rabbit (1:1500, Biorad) secondary antibodies were used. ImageJ (*Schindelin et al., 2012*) was used for quantification of western blots.

## Statistical analysis

Error bars represent the standard deviation of biological triplicates. Student's t-tests were performed using GraphPad Prism. Asterisks indicate statistically significant differences between mouse lines (*$p > 0.05$, **$p > 0.01$, ***$p > 0.005$).

## Illumina RNA-sequencing (human)

RNA-seq libraries were prepared with the Illumina TruSeq RNA Sample Preparation kit (Illumina) according to the manufacturer's protocol. Libraries were validated on an Agilent Bioanalyzer 2100. Indexed libraries were equimolarly pooled and sequenced on a SE50 (single-end 50 base pair) Illumina HiSeq2500 lane, which yielded an average of about $30 \times 10^6$ reads/sample.

## Illumina RNA-sequencing (mouse)

For generation of RNASequencing libraries, RNA was isolated as described above and the RNA integrity was tested with a BioAnalyzer (Agilent Technologies) or FragmentAnalyzer (Advanced Analytical). For RNASequencing, target RIN score of input RNA (500–1000 ng) usually had a minimum RIN score of 8. RNASequencing libraries were prepared with TruSeq stranded RNA sample preparation kits (Illumina), and depletion of ribosomal RNA was performed by positive selection of polyA+ RNA. Sequencing was performed on Illumina HighSeq or NextSeq machines. RNA-seq 50 bp reads were aligned to the mouse genome (assembly GRCm38/mm10) using TopHat (*Trapnell et al., 2009*). The Gencode M13 gtf was used as the input annotation. Differential gene expression-specific analyses were conducted with the DESeq (*Anders and Huber, 2010*) R package. Specifically, DESeq was used to normalize gene counts, calculate fold change in gene expression, estimate p-values and adjusted p-values for change in gene expression values, and perform a variance-stabilizing transformation on read counts to make them amenable to plotting. Data was submitted to GEO GSE141754.

## Gene expression analysis

DESeq2 v1.22.2 (*Love et al., 2014*) was used to create counts tables (*Supplementary files 6-7*) and complete differential gene expression analysis on RNA-seq data from human monocyte-derived macrophage ± 18 hr LPS and mouse BMDM ± 6 hr LPS experiments. The sample conditions used were 'control' and 'LPS.' Data was plotted using ggplot2 v3.1.1 (*Wickham, 2009*). Significance thresholds were set to |log2FC| ≥ 2 and adjusted p-value≤0.05. The list of genes with significant AFE events was then highlighted on the appropriate graphs.

## Alternative splicing quantification (PSI)

JuncBASE (*Brooks et al., 2011*) was used to identify and quantify alternative splicing events. After the identification of each alternative splicing event, JuncBASE counts reads supporting the inclusion and

exclusion isoform of each event. Isoform abundances are then calculated by dividing the read counts for the isoform by the length of the isoform. Ψ-values for each splicing event are derived from the isoform abundances:

PSI formula:

$$PSI = \text{Inclusion Isoform Abundance} / (\text{Inclusion Isoform Abundance} + \text{Exclusion Isoform Abundance})$$

## Nanopore direct RNA-sequencing

### Total RNA extraction

Total RNA was extracted according to *Workman et al., 2019*. $5 \times 10^7$ frozen macrophages were resuspended in 3 mL of TRI-Reagent (Invitrogen AM9738) and vortexed for 5 min. The mixture was incubated at RT for 5 min, transferred to 1.5 mL tubes, and spun down to remove debris. Supernatant was transferred to fresh tubes and chloroform extracted. The aqueous portion was mixed with an equal volume of isopropanol, incubated for 15 min at RT, and centrifuged at 12,000 g at 4°C. Pellet was washed twice with 75% ethanol, air dried, and resuspended in nuclease-free water.

### Poly(A) RNA isolation

100 μg aliquots of total RNA preparation were brought to 100 μL in nuclease-free water and poly-A selected using NEXTflex Poly(A) Beads (BIOO Scientific Cat#NOVA-512980) according to the manufacturer's instructions. The resulting poly-A RNA solution was stored at –80°C.

### Library preparation and MinION run

A native RNA-sequencing library was prepared following the ONT SQK-RNA001 using Superscript IV (Thermo Fisher) for the reverse transcriptase step. Sequencing was performed using ONT R9.4 flow cells and the standard MinKNOW protocol.

### Basecalling and sequence alignments

ONT albacore version 2.1.0 was used to baseball Nanopore direct RNA raw signal. We used minimap2 (*Li, 2016*) with default parameters to align reads to the mm10 mouse genome reference. Following alignment, we used SAMtools (*Li et al., 2009*) to filter out reads with mapping quality (MAPQ) less than 30.

## Alignment of paired-end mouse RNA-seq data

Bowtie2-build v2.3.1 (*Langmead and Salzberg, 2012*) was used to build the index files from GRCm38. p6 mouse (mm10 assembly) genome sequence obtained from Gencode. The index files were then used for completing paired-end alignment of each sample using TopHat2 v2.1.1 (*Kim et al., 2013*) with parameters: *--segment-length 20, --library-typ fr-firststrand, --no-coverage-search*.

## Identification of splicing events and calculating PSI

Human monocyte-derived macrophage ± LPS and mouse BMDM ± LPS were each run through Junc-BASE v1.2 (*Brooks et al., 2011*) to calculate PSI values and identify splicing events. The JuncBASE parameters used for the identification of splicing events and calculation of PSI in human monocyte-derived macrophage ± LPS are *-c 1.0 j [introns from Gencode v24* (hg19 assembly) (*Frankish et al., 2019*) *--jcn_seq_len 88*]. The JuncBASE parameters used for the identification of splicing events and calculation of PSI in mouse bone marrow-derived macrophage ± LPS are: *-c 1.0 j [introns from Gencode M18* (mm10 assembly) (*Frankish et al., 2019*) *--jcn_seq_len 88*].

## Differential splicing analysis

Differential splicing analysis was completed using DRIMSeq v1.10.1 (*Nowicka and Robinson, 2016*) and the compareSampleSets.py script within JuncBASE. CompareSampleSets.py applies the statistical t-test and DRIMSeq applies the framework of the Dirichlet-multinomial distribution for differential analysis. Each tool was used to apply the respective statistical method in order to determine significant differentially spliced events between control (-LPS) and LPS (+LPS) conditions. The AS_exclusion_inclusion_counts_lenNorm.txt JuncBASE output table from the identification and quantification analysis of each experiment was used as the input table for both compareSampleSets.py and DRIMSeq.

For all experiments, compareSampleSets.py was run using parameters: `--mt_correction BH --which_test t-test --thresh 10 --delta_thresh 5.0`. The following parameters were used for the differential splicing analysis of data from human monocyte-derived macrophage ± LPS with DRIMSeq: *min_samps_gene_expr = 8, min_samps_feature_expr = 4, min_gene_expr = 10, min_feature_expr = 0*. The following parameters were used for the differential splicing analysis of data from mouse BMDM ± LPS with DRIMSeq: *min_samps_gene_expr = 6, min_samps_feature_expr = 3, min_gene_expr = 10, min_feature_expr = 0*. Following differential splicing analysis using each tool, genes with significant differential splicing events were filtered for using thresholds of a corrected/adjusted p-value≤0.25 and a |Δ PSI| ≥ 10. Within each category of event type, the union of genes with significant events identified using compareSampleSets.py and DRIMSeq within each experiment was used for further comparison. Novel intron retention events, annotated with a 'N,' were removed for further analyses.

### Ranking analysis

Following differential splicing analysis with JuncBASE, events were ordered by p-value. A subset of the top 5, 10, 15, 20, and 25% events were quantified by event type. Only events having a |Δ PSI| ≥ 10 were considered. The same analysis was completed following differential splicing analysis with DRIMSeq2.

### Analysis of *Listeria* and *Salmonella* data sets following 24 hr stimulation

Data for the 24 hr time point was downloaded from GEO (GSE73502) for control and experimental conditions: *Listeria* and *Salmonella* (*Pai et al., 2016*). JuncBASE and differential alternative splicing analysis was run using each pair of control and experimental samples and parameters as described above. Following differential alternative splicing analysis, alternative splicing events were categorized by event type using two significance thresholds: |Δ PSI| ≥ 10 and adjusted p-value≤0.05 or |Δ PSI| ≥ 10 and adjusted p-value≤0.25. Only known (K) events were considered for intron retention events. Jcn_only_AD and jcn_only_AA events were not considered.

### Identification of high-confidence isoforms from nanopore data

FLAIR (*Tang et al., 2020*) was used to assemble the high-confidence isoforms from native RNA sequencing of mouse BMDM ± 6 hr LPS. FLAIR modules align, correct, and collapse were used for the assembly. Corresponding short-read data was used when running the correct module in order to help increase splice-site accuracy. Putative promoter regions were obtained using ATAC-seq data from *Atianand et al., 2016*; *Tong et al., 2016* converted to mm10 coordinates using liftOver (*Hinrichs et al., 2006*), and used when running the collapse module.

### Creating merged reference annotation files and incorporating nanopore

The *isoforms.gtf* output file from FLAIR collapse was combined with the Gencode M18 (mm10 assembly) basic annotation using cuffmerge from Cufflinks v2.2.1 (*Trapnell et al., 2010*) with parameter: -s GRCm38.p6.genome.fa. Similarly, the *isoforms.gtf* output file was combined with the Gencode M18 (mm10 assembly) comprehensive annotation with parameter: -s GRCm38.p6.genome.fa. The resulting comprehensive annotation output file was used to generate an intron coordinate file for the identification of splicing events and calculating PSI of splicing events found in mouse BMDM ± 6 hr LPS using JuncBASE with parameters: *-c 1.0, -j [intron coordinates from merged comprehensive annotation], --jcn_seq_len 238*. Parameters used for finding significantly differentially spliced events using compareSampleSets.py from JuncBASE are: *--mt_correction BH --which_test t-test --thresh 10 --delta_thresh 5.0*. Parameters used for finding significantly differentially spliced events using DRIMSeq are: *min_samps_gene_expr = 6, min_samps_feature_expr = 3, min_gene_expr = 10, min_feature_expr = 0*.

### Creating and comparing gene lists

For each experiment, a table with the union of significant events found using DRIMSeq and compareSampleSets.py was created. A list of genes with significant events was generated for each experiment using this table. BioVenn (*Hulsen et al., 2008*) and DrawVenn (*Li, 2016*) were then used to remove

duplicate gene names and compare the lists of genes to find unique and common genes between experiments.

## Identification of genes with conserved alternative splicing events

Following differential splicing analysis with JuncBASE and DRIMSeq2, gene names of significant events (|Δ PSI| ≥ 10 and adjusted p-value≤0.25) were curated by event type. Overlaps were determined between human and mouse JuncBASE and DRIMSeq2 events determined with and without the support of Nanopore-identified transcripts.

## Comparison of all alternative splicing events identified in mouse using Gencode or de novo + Gencode annotations

Following differential splicing analysis with JuncBASE and DRIMSeq2, gene names of significant events (|Δ PSI| ≥ 10 and adjusted p-value≤0.25) were curated by event type. Overlaps were determined between mouse JuncBASE and DRIMSeq2 events determined with and without the support of Nanopore-identified transcripts.

## Validation of FLAIR-identified TSS with CAGE data

Coordinates corresponding to mouse CAGE peaks (mm9.cage_peak_phase1and2combined_coord. bed) were downloaded from the FANTOM5 database (*FANTOM Consortium and the RIKEN PMI and CLST (DGT), 2014*). Coordinates were converted from mm9 to mm10 using liftOver. TSS of FLAIR-identified isoforms were extracted using the pull_starts.py script included in the FLAIR software. TSS were extracted from the Gencode M18 annotation using a custom script and used as the set of known TSS. TSS of FLAIR-identified isoforms were annotated as known if they overlapped with known TSS using bedtools (v 2.25.0) tool intersectBed. TSS of FLAIR-identified isoforms were annotated as being novel if they did not overlap with any known TSS. These were also identified using bedtools tool intersectBed with parameter: -v. CAGE peaks that overlapped with known and novel FLAIR-identified TSSs were identified using bedtools tool intersectBed, and the distribution of CAGE scores was plotted for each group.

Coordinates corresponding to CAGE peaks identified as true TSS by TSS classifier (TSS_mouse. bed) were downloaded from the FANTOM 5 database. Coordinates were converted from mm9 to mm10 using liftOver. The proportion of true known and novel FLAIR-identified TSS sites was identified by determining TSSs that overlapped with the CAGE peaks. Bedtools (v 2.25.0) tool intersect was used to complete this analysis.

## Differential chromatin accessibility

Raw ATAC-seq fastq sequence files were published in *Tong et al., 2016* and pulled from the GEO accession number GSE67357. A bowtie2 (*Langmead and Salzberg, 2012*) index file was created from the GENCODE mm10 version M18 genome annotation file, and the untreated and LPS-treated ATAC-seq reads were aligned using the created index file with the default bowtie2 parameters. Peaks were then called separately by treatment type on untreated and treated samples using the ENCODE published ATAC-seq peak calling pipeline (*ENCODE Project Consortium, 2012*, *Lee et al., 2021a*, https://github.com/ENCODE-DCC/atac-seq-pipeline) using the aligned reads as sequence input. Parameters that were used followed the basic JSON input file template, using an IDR threshold of 0.05. Peaks from both conditions were then merged using bedtools merge (*Quinlan and Hall, 2010*) if the tail ends were less than 10 bp away from each other in order to create a set of consensus peaks from both conditions. A GFF file was created from the merged peaks, assigning a unique ID to each peak. This GFF file was provided to HTSeq-count (*Anders et al., 2015*) along with the aligned reads for each condition in each replicate to count reads aligning to the unique peaks. The read count matrix was provided to DESeq2 (*Love et al., 2014*) to call differential peaks. All peaks were considered significant if log2FC ≥ 0.8 and p-value≤0.15.

## Differential transcription factor binding

ChIP-seq fastq sequencing files for the NF-κB subunit p65 and interferon transcription factor Irf3 were downloaded from the GEO accession number GSE67357 published by *Tong et al., 2016*. ChIP-seq samples for input control, untreated, and LPS treatment were aligned using bowtie2 (*Langmead

*and Salzberg, 2012*) to the mm10 version M18 mouse genome annotation with default parameters. Peaks were separately called between untreated and treated conditions using the ENCODE published ChIP-seq peak calling pipeline (*ENCODE Project Consortium, 2012*, *Lee et al., 2021b*, https:// github.com/ENCODE-DCC/chip-seq-pipeline2) from the aligned reads. The aligned input control reads were input as genomic background to account for noise in ChIP-seq experiments. The basic JSON input template file was used using an IDR threshold of 0.05. Differential peak analysis was done using the HOMER suite designed for ChIP-seq data (*Heinz et al., 2010*). Consensus peaks from both conditions were merged using mergePeaks within HOMER, reporting the direct overlap between peaks. Tag directories were created to count reads for each aligned sequence file with TagDirectory. The merged consensus peaks were then annotated for raw read counts using the tag directories for each replicate and condition with the annotatePeaks.pl tool. Annotated consensus peaks were provided to getDiffExpression.pl, normalizing to total read counts. Peaks were considered significant if they had a corrected p-value$\leq$0.25 and log2FC $\geq$ 1.

## Alternative splicing event overlap

To identify differential TF binding and chromatin remodeling at the promoters of the observed AFE events, the coordinates of the alternate first exon were determined from the statistical testing results. Significant (p-value$\leq$0.05) AFE events were first filtered out from all results. For all significant results, if the inclusion exon had a $\Delta$ PSI $\geq$10, the inclusion exon coordinates from the JuncBASE table (*Supplementary files 1-3*) were used as the coordinates for that splicing event. If the inclusion exon had a $\Delta$ PSI $\leq$ –10, all other inclusion exons for that splicing event from the statistical testing (DRIMSeq or t-test) were considered, and any inclusion exon with $\Delta$ PSI >10 was used. Redundant events with the same exon coordinates were then filtered out, leading to a final set of exon coordinates. The coordinates were then extended to include 10 kbp upstream of the exon. Overlap of differential chromatin accessibility and different TF binding was done using bedtools intersect (*Quinlan and Hall, 2010*) with the significant differential peak coordinates and the AFE 10 kbp upstream region, returning the coordinates of the exon that show differential chromatin accessibility or TF binding.

## HOMER promoter analysis

HOMER (*Boeva, 2016*) was used to search for sequence motifs in promoter regions of LPS-specific AFEs. The start coordinates from the BED file were extended to include the 3 kb upstream of the exon. These extended regions were used as input for HOMER's findMotifsGenome.pl script, along with the mm10 reference genome from within HOMER and the size parameter set to 'given.' findMotifsGenome.pl was run initially with the -preparse flag to parse the reference genome based on the size of the input sequences and then was ran after without the preparse flag in order to generate the motif output file (*Supplementary files 10-11*).

## BED file generation of inclusion and exclusion AFE

Significant (corrected p-value$\leq$0.25) AFE events were first filtered out from all results. For all significant results, if the inclusion exon had a $\Delta$ PSI $\geq$10, the inclusion exon coordinates from the JuncBASE table were used as the coordinates for that splicing event. If the inclusion exon had a $\Delta$ PSI $\leq$ –10, all other exons for that splicing event from the statistical testing (DRIMSeq or t-test) were considered, and any other exon with $\Delta$ PSI $\geq$10 was used (*Supplementary files 12-13*).

## Acknowledgements

We thank Eric Martin for sharing his time and knowledge on HOMER. We would also like to thank Alison Tang for sharing her knowledge and assisting with FLAIR isoform analysis. We also thank Dr. Vijay Rathinam and Dr. Kate Fitzgerald for their critical insights into this manuscript. Finally, we would like to thank Kevin S Johnson and Dr. Karen Ottemann who provided reagents for the IRE mutagenesis experiments. Funding for this work was partially supported by a Special Research Grant/Collaborative Research Grant from the UCSC Committee on Research (COR) to ANB and SCa. Additional funding support from NIH HG010053 (ANB and MA) and Oxford Nanopore Research Grant SC20130149 (MA).

## Additional information

### Competing interests

Mark Akeson: holds options in Oxford Nanopore Technologies (ONT), is a paid consultant to ONT, received reimbursement for travel, accommodation and conference fees to speak at events organized by ONT,received research funding from ONT and is an inventor on 11 UC patents licensed to ONT (6,267,872, 6,465,193, 6,746,594, 6,936,433, 7,060,50, 8,500,982, 8,679,747, 9,481,908, 9,797,013, 10,059,988, and 10,081,835). Angela N Brooks: received reimbursement for travel, accommodation and conference fees to speak at events organized by Oxford Nanopore Technologies (ONT). The other authors declare that no competing interests exist.

### Funding

| Funder | Grant reference number | Author |
| --- | --- | --- |
| NIH Office of the Director | HG010053 | Angela N Brooks<br>Mark Akeson |
| Oxford Nanopore Technologies | SC20130149 | Mark Akeson |
| University of California, Santa Cruz | Committee on Research | Angela N Brooks<br>Susan Carpenter |

The funders had no role in study design, data collection and interpretation, or the decision to submit the work for publication.

### Author contributions

Elektra K Robinson, Conceptualization, Data curation, Formal analysis, Methodology, Validation, Visualization, Writing – original draft, Writing – review and editing; Pratibha Jagannatha, Conceptualization, Formal analysis, Methodology, software, Visualization, Writing – review and editing; Sergio Covarrubias, Conceptualization, Methodology; Matthew Cattle, Methodology, software, Visualization; Valeriya Smaliy, Validation; Rojin Safavi, Formal analysis, Investigation; Barbara Shapleigh, Miten Jain, Methodology; Robin Abu-Shumays, Methodology, Writing – review and editing; Suzanne M Cloonan, supervision, Writing – review and editing; Mark Akeson, Funding acquisition, Resources; Angela N Brooks, Conceptualization, Funding acquisition, Investigation, Methodology, Project administration, Resources, Writing – review and editing; Susan Carpenter, Funding acquisition, Project administration, supervision, Writing – review and editing

### Author ORCIDs

Elektra K Robinson ⓘ http://orcid.org/0000-0003-1427-3897
Pratibha Jagannatha ⓘ http://orcid.org/0000-0003-0257-3139
Angela N Brooks ⓘ http://orcid.org/0000-0002-7898-3073
Susan Carpenter ⓘ http://orcid.org/0000-0002-5600-5404

### Ethics

All animal work was carried out in strict accordance with the recommendations in the Guide for the Care and Use of Laboratory Animals of the National Institutes of Health. The protocol was approved by the Institutional Animal Care and Use Committee at the University of California Santa Cruz (Protocol number CARPS1810).

### Decision letter and Author response

Decision letter https://doi.org/10.7554/eLife.69431.sa1
Author response https://doi.org/10.7554/eLife.69431.sa2

## Additional files

### Supplementary files

• Supplementary file 1. Numerical source data from human macrophages ± TLR4 RNA-seq analyzed by JuncBASE using GENCODE transcriptome for *Figure 1B*.

- Supplementary file 2. Numerical source data from mouse macrophages +/- TLR4 RNAseq analyzed by JuncBASE using GENCODE transcriptome for *Figure 1B and C*.
- Supplementary file 3. Numerical source data from mouse macrophages ± TLR4 RNA-seq analyzed by JuncBASE using GENCODE+ de novo transcriptome for *Figure 1C*.
- Supplementary file 4. Numerical source data from human macrophages ± *Listeria* RNA-seq analyzed by JuncBASE using GENCODE transcriptome for *Figure 1—figure supplement 4*.
- Supplementary file 5. Numerical source data from human macrophages ± *Salmonella* RNA-seq analyzed by JuncBASE using GENCODE transcriptome for *Figure 1—figure supplement 4*.
- Supplementary file 6. Numerical source data from human macrophages ± TLR4 RNA-seq analyzed by DESeq2 using GENCODE transcriptome for *Figure 1E*.
- Supplementary file 7. Numerical source data from mouse macrophages ± TLR4 RNA-seq analyzed by DESeq2 using GENCODE transcriptome for *Figure 1F*.
- Supplementary file 8. Numerical source data from mouse macrophages ± TLR4 ATAC-seq analyzed using a peak calling pipeline for *Figure 3A*.
- Supplementary file 9. Numerical source data from mouse macrophages ± TLR4 p65 and IRF3 ChIP-seq analyzed using a peak calling pipeline for *Figure 3A*.
- Supplementary file 10. Numerical source data from all alternative first exon promoters analyzed by HOMER for *Figure 3B*.
- Supplementary file 11. Numerical source data from unannotated and annotated Aim2 promoter analyzed by HOMER for *Figure 3C*.
- Supplementary file 12. Numerical source data of inclusion alternative first exon coordinates for *Figure 4—figure supplement 1A*.
- Supplementary file 13. Numerical source data of exclusion alternative first exon coordinates for *Figure 4—figure supplement 1A*.
- Supplementary file 14. Numerical source data of unannotated and annotated Aim2 5'UTR analyzed by RegRNA2.0 for *Figure 4F*.
- Transparent reporting form

## Data availability

Sequencing data have been deposited in GEO under accession codes GSE141754.

The following dataset was generated:

| Author(s) | Year | Dataset title | Dataset URL | Database and Identifier |
|---|---|---|---|---|
| Cattle M, Robinson EK, Jagannatha P, Covarrubias S | 2021 | Inflammation drives alternative first exon usage of critical immune genes including Aim2 | https://www.ncbi.nlm.nih.gov/geo/query/acc.cgi?acc=GSE141754 | NCBI Gene Expression Omnibus, GSE141754 |

The following previously published datasets were used:

| Author(s) | Year | Dataset title | Dataset URL | Database and Identifier |
|---|---|---|---|---|
| Pai AA, Baharian G, Pagé Sabourin GA, Nédélec GY, Grenier GJ, Siddle GKJ, Dumaine GA, Yotova GV, Burge GCB, Barreiro GLB | 2016 | Widespread shortening of 3' untranslated regions and increased exon inclusion characterize the human macrophage response to infection | https://www.ncbi.nlm.nih.gov/geo/query/acc.cgi?acc=GSE73502 | NCBI Gene Expression Omnibus, GSE73502 |
| Tong AJ, Liu X, Thomas BJ, Lissner MM | 2016 | A Stringent Systems Approach Uncovers Gene-Specific Mechanisms Regulating Inflammation | https://www.ncbi.nlm.nih.gov/geo/query/acc.cgi?acc=GSE67357 | NCBI Gene Expression Omnibus, GSE67357 |

*Continued on next page*

*Continued*

| Author(s) | Year | Dataset title | Dataset URL | Database and Identifier |
|---|---|---|---|---|
| Tong A, Thomas BJ | 2016 | A Stringent Systems Approach Uncovers Gene-Specific Mechanisms Regulating Inflammation | https://www.ncbi.nlm.nih.gov/geo/query/acc.cgi?acc=GSE74191 | NCBI Gene Expression Omnibus, GSE74191 |
| Link VM, Glass CK | 2018 | Analysis of genetically diverse macrophages reveals local and domain-wide mechanisms that control transcription factor binding and function | https://www.ncbi.nlm.nih.gov/geo/query/acc.cgi?acc=GSE109965 | NCBI Gene Expression Omnibus, GSE109965 |
| Tong AJ, Liu X, Thomas BJ, Lissner MM | 2016 | A Stringent Systems Approach Uncovers Gene-Specific Mechanisms Regulating Inflammation | https://www.ncbi.nlm.nih.gov/geo/query/acc.cgi | NCBI Gene Expression Omnibus, GSE67343 |
| Song R, Dozmorov I, Malladi V, Liang C, Arana C, Wakeland B, Pasare C, Wakeland EK | 2021 | IRF1 governs the differential Interferon-Stimulated Gene responses in human monocytes and macrophages by regulating chromatin accessibility | https://www.ncbi.nlm.nih.gov/geo/query/acc.cgi?acc=GSE147310 | NCBI Gene Expression Omnibus, GSE147310 |
| The FANTOM Consortium and the RIKEN PMI and CLST (DGT) | 2014 | A promoter-level mammalian expression atlas | https://ddbj.nig.ac.jp/DRASearch/submission?acc=DRA000991 | DDBJ, DRA000991 |

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
