## [Decision Letter]

[Editors’ note: the authors submitted for reconsideration following the decision after peer review. What follows is the decision letter after the first round of review.]

Thank you for submitting your work entitled "Inflammation Drives Alternative First Exon usage to Regulate Immune Genes including a New Iron Regulated Isoform of Aim2" for consideration by *eLife*. Your article has been reviewed by 3 peer reviewers, and the evaluation has been overseen by a Reviewing Editor and a Senior Editor. The reviewers have opted to remain anonymous.

Our decision has been reached after consultation between the reviewers. Based on these discussions and the individual reviews below, we regret to inform you that your work cannot be considered further for publication in *eLife* at this time.

There was significant enthusiasm for the work. However, it seems that considerable effort including additional experiments will be required to firm up the conclusions and make the paper suitable for *eLife*. Accordingly we must reject the paper in its current form. Nevertheless, we encourage you to resubmit if and when you can address the majority of the reviewers' concerns. In this regard, addressing both functional relevance and mechanism would be ideal, but addressing one of these topics will be sufficient.

*Reviewer #2:*

Our understanding of the transcriptomic impact of innate immune signaling remains incomplete. Here Robinson et al., use both long and short read RNA sequencing to gain further insight into LPS-induced changes to mRNA isoform expression in human and mouse macrophages. Their studies report the novel observation that the most common change in isoform expression is alternative use of the first exon. Such changes are indicative of transcriptional regulation, and is thus consistent with the known impact of innate immune signaling on activation of multiple transcription factors. Despite some minor concerns with details of the study, this is a well-executed and important study that will be of interest and importance to many studying innate immunity, as well as those interested in gene regulation.

1. In some ways this is minor, but the authors should be careful to not describe alternative first exon use as alternative splicing. While a novel splice junction is created, mechanistically this is driven by changing transcriptional regulation, and then splicing occurs in the only pattern available to that TSS. In general this is described appropriately in the manuscript, but at a few points there is confusing terminology.

2. An interesting and somewhat surprising point in the manuscript is that 50% of the AFE events don't show an overall change in gene expression. For Aim2, which does change, the authors show that the AFE change is due to activated use of the unannotated TSS in LPS-stimulated cells. For those genes for which AFE use doesn't correlate with a change in gene expression (e.g. Ncoa7, Rcan1, Ampd3 – Figure S3) is there still transcriptional activation of one TSS and transcriptional silencing of the other? In other words, is there coordinated regulation of the two TSSs to ensure overall message abundance doesn't change, or does activation of one TSS inherently shut off the other (more akin to splice site competition in traditional AS)?

3. The data suggesting that an IRE regulates translation of the induced 5'UTR is compelling, but more work should be done to confirm. Most importantly, the experiment in Figure 4J should be repeated with the deltaIRE version of the unannotated UTR. Also is the IRE regulation controlled upon LPS-stimulation, or just the presence of the IRE element? In other words, what is the distribution of the annotated and unannotated isoforms in the polysome in the absence of LPS (i.e. repeat 4P without LPS)? Can the authors comment on whether the level of iron or the activity of IRP1/2 change in LPS-stimulated cells?

*Reviewer #3:*

This manuscript by Robinson et al. presents an interesting and timely analysis of a wealth of transcriptome data upon immune stimulation. The unique combination of long-read Oxford Nanopore and short-read Illumina high-throughput sequencing across both human and mouse samples presents an opportunity many interesting inter-species immune response comparisons, as well as elucidation of full-length transcript information. This paper is well-written and has interesting validation and discussions regarding Aim2. My major concern is that the paper seems to narrow in on the characterization of Aim2 and class of RNA processing changes (alternative first exons) quite quickly without really delving into the rest of the data and how they arrived there. Below are my major comments and suggestions:

1. I would have liked the authors to provide more insight into how they honed-in on specifically talking about first exon changes, by discussing more of the other RNA processing changes they found. There is cursory mention in the text and figures of other alternative exon or splice site changes. Firstly, other studies (including those referenced by the authors) have found hundreds RNA processing changes genome-wide upon immune stimulation – especially of cassette exons, alternative splice sites, and last exon/3'UTR changes. However here, the authors only find tens of changes (Figure 1B). Are they underpowered to identify changes and can they do any sort of analyses to show that they are sufficiently powered (# of sequencing reads and junctions, complexity of reads, etc)?

2. Similarly, I would also be interested in seeing an analysis indicating whether the 50 AFE events that overlap between the long-read and short-read sequencing analyses is a statistically significant overlap. Particularly, how many overlapping events would be expected given the difference in quantification power between the two methods? How many real AFE differences might the authors be missing because the long-read sequencing methods often do not have the power to identify them (ie. lower expressed genes in one or the other condition, thus dropout of isoforms and perhaps fewer isoform differences for differentially expressed genes).

3. Second, for the non-AFE changes that they did find, there is very little discussion about what those changes might represent. Specifically: (a) how many changes are validated with long-read data?, (b) is there any insight into specific domains being included/changed, especially using the long-read data?, (c) how many of these non-AFE changes overlap between species? and (d) which types of genes show higher overlap between species and what are their characteristics (binding sites, etc)? To my knowledge, this is the first study that is really designed to properly really look at the *conservation* of splicing or RNA processing changes after immune activation, so I would love to see more analysis and discussion of this aspect genome-wide.

4. The authors define significant splicing changes as those with a p-value <= 0.25 and |dPSI| >= 10. I'd like some more clarification on whether this is an adjusted p-value (BH, FDR, or some other multiple test-corrected p-value). Especially if this is adjusted, I find it surprising that the authors are choosing such a liberal statistical confidence level and that even with such a liberal threshold, they are only getting tens of significant events. I would like the authors to at least show these same trends across multiple p-value thresholds or with rank threshold analysis (top 5%, top 10%, top 20%) to show biological trends.

5. The authors introduce their long-read sequencing data by mentioning that they wanted to identify "additional splicing events that are not captured using short-read sequencing." They then go onto to only talk about novel first exon events identified with the long-read sequencing data. Did they identify any other non-AFE events in using the long-read that could then be quantified with the short read data? And second, how do they quantify confidence for novel AFE isoforms, when long-read data seems to have lots of issues with properly sequencing the terminal ends of transcripts (particularly the 5' end when polyA primed, as occurs in ONT DirectRNA sequencing)? They mention the use of ATAC-seq data to show putative promoter support, but mention at one point in their methods that ATAC regions within 10kb of AFEs are considered. These seems like it could be a rather large region to be sure that the ATAC peak is specific to a novel AFE – what is the average distance between AFEs? Finally, I would love to also see the incorporation of CAGE-seq data (or other 5'end data) to validate the specific AFEs sites – which I believe the FANTOM consortium has across many human and mouse tissues.

*Reviewer #4:*

In this manuscript, Robinson et al., identified alternative first exon (AFE) switching events conserved between mouse and human following macrophage inflammation. Using short and long-read sequencing, the authors identified a few unannotated transcription initiation sites (TSS) that are specific of an inflammatory response. Among those, they centered on an unannotated TSS in the Aim2 gene that drives expression of a novel isoform regulated by an iron-responsive element in its 5′UTR.

While previous work had documented crucial AFE switching events in many other biological contexts, Robinson et al. presents here an interesting AFE switching event that can have potential implications for our understanding of the molecular regulation of the innate immune response. For publication in *eLife*, I would expect further progress on global mechanisms and biological relevance of these AFE switching events, as well as evidence that the AFE are truly first exons/TSSs.

1. Are the AFEs truly first exons/TSS? While both short-read and long-read sequencing detected changes in alternative splicing choices, neither of those are optimal methodologies to analyze first exons. Therefore, I suggest to use a more specialized method to identify (and quantify) more accurately the usage of first exons. Globally, cap analysis of gene expression (CAGE) would be ideal. For validation of specific AFE changes, the qPCR technique has a few issues. First, it does not have nucleotide resolution, so the authors should not refer to TSSs if they used this technique for validation. Second, many downstream first exons are also used as internal exons in other isoforms. There is not a direct technology to analyze specifically first exons/TSSs here. Also, RNA-sequencing technologies, depending on their depth, can definitely miss specific isoforms. Considering a low coverage in 5'end of genes in RNA-seq analysis, this is particularly important for first exons. A qPCR would only analyze the well-known TSSs. Thus, 5'RACE or a similar technology should be perform to assess the relative usage of AFE specifically.

2. Global mechanism. The authors assumed that the mechanism of AFE switching is generated by transcription initiation and looked for transcription factors binding and chromatin structure modifications in promoters. However, they did not ruled out the possibility that the global switching effect is a post-transcriptional regulation, such as differential mRNA stability. A transcription initiation measurement (e.g., 4SU metabolic labelling) is necessary to demonstrate that the changes in AFE usage are co-transcriptional. In addition, in terms of their ATAC-Seq analysis, the chromatin structure changes in promoters can be cause or consequence of transcription initiation. Thus, it should not be listed as one mechanism driving the expression of AFE events (line 145). Also, to demonstrate a mechanism based on transcription factor binding more than 2 transcription factors should be considered. In any case, the expression patterns of the transcription factors considered are not clear. As a minor note, the bioinformatic analysis of the two promoters regions driving the isoforms of Aim2 (line 156) is not explained in the method section.

3. Biological relevance. Could the authors evaluate whether the translation regulation of Aim2 based on its AFE switching is a more generalized phenomenon? Are there any global gene regulation changes triggered by the other genes with significant changes is AFE usage?

[Editors’ note: further revisions were suggested prior to acceptance, as described below.]

Congratulations, we are pleased to inform you that your article, "Inflammation Drives Alternative First Exon usage to Regulate Immune Genes including a Novel Iron Regulated Isoform of Aim2", has been accepted for publication in *eLife*. Your revised article has been reviewed by 3 peer reviewers, and the evaluation has been overseen by a Reviewing Editor and a Senior Editor. The reviewers have opted to remain anonymous.

*Reviewer #1:*

I appreciate the authors' efforts to address the issues of global mechanism and biological relevance, as well as other points raised. This new version of the manuscript is significantly stronger and it represents an important contribution to the literature. However, I still have concerns about the evidence the authors present to demonstrate that the AFE are "true" first exons/TSSs.

In my previous report, I suggested the authors use CAGE instead of RNA-seq (either short or long-reads) and 5'RACE instead of qPCR, both specific techniques to analyze first exons/TSSs. In this version, the authors used CAGE data from the FANTOM project to classify their first exon calls as "true" TSS. They found that only 45% of the novel TSS identified overlap with CAGE peaks. In my opinion, this raise the possibility that the other 55% are not "true" TSSs. Further, before publication, I would expect 5'RACE or any other direct method to validate that specific TSS are "true" TSS, at least for Denr, Arhgef7 and Aim2 which they validated using RT-qPCR.

*Reviewer #2:*

1. While the authors have now included a brief discussion about whether AFEs are regulated by transcription, splicing, or both in their discussion, I still think the authors should tone down the mechanistic implications of the language throughout the rest of the text, perhaps use the term "alternative RNA processing changes" instead of "alternative splicing changes"?

2. The authors mention that they implemented a new parameter in their FLAIR analysis to incorporate a bed file of ATAC annotations to calibrate the identification of novel AFEs. Do the authors plan to release this new parameter as an update to their existing software? I think this would be of great use to the community that analyzes long-read RNA data.

*Reviewer #3:*

The authors have fully addressed all of the concerns raised by adding additional data. This enhances the significance of the work and strengthens the conclusions.

---

## [Author Response]

[Editors’ note: the authors resubmitted a revised version of the paper for consideration. What follows is the authors’ response to the first round of review.]

Reviewer #2:Our understanding of the transcriptomic impact of innate immune signaling remains incomplete. Here Robinson et al., use both long and short read RNA sequencing to gain further insight into LPS-induced changes to mRNA isoform expression in human and mouse macrophages. Their studies report the novel observation that the most common change in isoform expression is alternative use of the first exon. Such changes are indicative of transcriptional regulation, and is thus consistent with the known impact of innate immune signaling on activation of multiple transcription factors. Despite some minor concerns with details of the study, this is a well-executed and important study that will be of interest and importance to many studying innate immunity, as well as those interested in gene regulation.1. In some ways this is minor, but the authors should be careful to not describe alternative first exon use as alternative splicing. While a novel splice junction is created, mechanistically this is driven by changing transcriptional regulation, and then splicing occurs in the only pattern available to that TSS. In general this is described appropriately in the manuscript, but at a few points there is confusing terminology.

Many thanks for this discussion point. There are a couple of nuances that we would like to address and that we hope clarifies our use of language throughout this manuscript. We now clearly define that AFEs involve alternative TSS usage coupled with alternative splicing and thus changed the terminology to “Alternative Isoform Usage”, where more appropriate. Not all of the AFE events are only changed due to transcriptional regulation. In the literature, we do see evidence of alternative splicing coupled with a new TSS and upstream exon. A recent study by Fiszbein et al. challenges the notion that promoter choice is first, but rather the splicing of internal exons can impact promoter choice (PMID: 31787377). This study shows that internal exons can dictate which first exon is spliced in the mature RNA molecule. We hope that using “Alternative Isoform Usage” strikes a balance that is clearer in this version of the manuscript.

2. An interesting and somewhat surprising point in the manuscript is that 50% of the AFE events don't show an overall change in gene expression. For Aim2, which does change, the authors show that the AFE change is due to activated use of the unannotated TSS in LPS-stimulated cells. For those genes for which AFE use doesn't correlate with a change in gene expression (e.g. Ncoa7, Rcan1, Ampd3 – Figure S3) is there still transcriptional activation of one TSS and transcriptional silencing of the other? In other words, is there coordinated regulation of the two TSSs to ensure overall message abundance doesn't change, or does activation of one TSS inherently shut off the other (more akin to splice site competition in traditional AS)?

Thank you for your interest in the other AFE genes. We have now included further characterization information of inclusion and exclusion isoforms of Ncoa7, Rcan1, Ampd3, Denr and Arhgef7 in supplemental figure 11. We show that of the 5 candidates, only Denr is significantly differentially expressed, with a log2 fold change more than 2 or -2. While Ncoa7, Rcan1, Ampd3 and Arhgef7 are not significantly differentially expressed with a log2FC more than 2. However, more importantly, these normalized counts show that all genes are expressed with or without LPS stimulation. Furthermore, looking at figure 1 (G-H) it is clear that Arhgef7 exclusion TSS is shut off by LPS, while both the Denr isoforms, at low amounts, are expressed with or without LPS.

Supplemental figure 11G shows that Ncoa7 inclusion TSS isoform is solely expressed with LPS, 11H shows that Rcan1 exclusion TSS closes with LPS, and 11I shows Ampd3 has both isoforms expressed with and without LPS.

Now to take a step further to determine the potential mechanism of how the promoters for these AFE are regulated, we have utilized publicly available tracks tracks including ATACseq (GSE74191), PU.1 ChIPseq (GSE109965), p65 ChIPseq (GSE67343) and IRF3 ChIP seq (GSE67343). In supplemental figure 11F for Ncoa7, we cannot conclude a mechanism driving from inflammatory TFs since there are no p65 or IRF3 binding driving the inclusion isoform and the PU.1, myeloid specific transcription factor is most enriched to the exclusion isoform. Interestingly, The ATAC peak for the inclusion isoform for Ncoa7 is more open for the LPS BMDM sample, which could conclude that the promoter is opened by LPS. Rcan1, SupFig11H, shows ATAC opening by inflammation for the inclusion isoform, this promoter also shows enrichment of IRF3 for the inclusion isoform promoter. SupFig11I for Ampd3, it is clear that the inclusion isoform is driven by p65 and IRF3 binding, however the promoter of the exclusion is less clear since the ATACseq shows the promoter to be closed. Arhgef7, SupFig11J, shows a clear strong promoter for the novel inclusion isoform. This is shown by the strong PU.1 binding, ATACseq reads, as well as IRF3 and p65 binding. The exclusion isoform is open as shown by ATACseq reads and has PU.1 binding, but much less in comparison to the inclusion isoform. Finally, SupFig11K for Denr, shows a convoluted promoter region. These two isoforms TSSs are about 250bp away from one another, therefore the ATACseq reads, p65 and IRF3 binding are not distinct between promoters, but to give confidence to the promoter regions it is clear that there are unique promoters by examining the PU.1 binding in BMDMs +/-LPS. In conclusion the mechanisms governing TSS choice and isoform usage are complex and gene and context specific. (Manuscript Lines 199-201, 209-212, 363-366)

3. The data suggesting that an IRE regulates translation of the induced 5'UTR is compelling, but more work should be done to confirm. Most importantly, the experiment in Figure 4J should be repeated with the deltaIRE version of the unannotated UTR.

Thank you for the suggestion of this important experiment. We have generated a ΔIRE BMDM cell line data in supplemental figure 15, which supports our findings and strengthens our new mechanism of posttranscriptional regulation of Aim2. To generate a ΔIRE we used our lab’s previously published CRISPR/Cas9 immortalized BMDM cell line (PMID: 29051223) and gRNAs were chosen based off of the CRISPR/Cas9 target tracks on the genome browser (PMID: 27380939, https://genome.ucsc.edu/cgi-bin/hgTrackUi?hgsid=1041360567_lXenOcRdA3sNMjAw8PODB1MiRgWj&c=chr1&g=crisp).

After careful selection of 3 gRNAs, we electroporated them in our BMDM Cas9 cell line, and through immediate cellular dilution we generated a KO IRE cell line. Performing LPS time course experiments, similar to that of Fig4R-S, we determine that the Aim2 protein is regulated by this IRE motif. Our WT BMDM Cas9 cell line shows a loss of protein at the 48 hr LPS time point while our ΔIRE BMDM Cas9 cell line shows a significant increase in Aim2 protein at this time point. With these novel cell lines and data, we are confident in our proposed IRE mechanism of regulating

Aim2. (Manuscript Lines 278-288, 384-390).

Also is the IRE regulation controlled upon LPS-stimulation, or just the presence of the IRE element? In other words, what is the distribution of the annotated and unannotated isoforms in the polysome in the absence of LPS (i.e. repeat 4P without LPS)?

The post-transcriptional regulation is completely controlled by the IRE hairpin present in the novel isoform of Aim2. Figure 2D-H, clearly shows that the annotated Aim2 isoform expression is not regulated by inflammation, while the IRE containing unannotated isoform is highly upregulated during inflammation. Additionally, we have generated a new cell line that lacks the IRE hairpin, genomically, and the Aim2 protein is no longer regulated by LPS, shown in supplemental figure 15. Finally, in figure 4Q, we redid the polysome profiling experiment to include unstimulated macrophages. In these data, we see that the annotated transcript is still enriched in the high polysome fraction and the unannotated transcript is enriched in the low polysome fraction, and these data are not dependent on LPS. (Manuscript Lines 257-266)

Can the authors comment on whether the level of iron or the activity of IRP1/2 change in LPS-stimulated cells?

We have now added the expression of IRP1 and IRP2, found in supplemental figure 14, showing that they are not significantly differentially expressed by LPS in primary BMDMs. Therefore, the amount of IRP1 or IRP2 would likely not be affected by LPS. (Manuscript Lines 265-268).

Reviewer #3:This manuscript by Robinson et al. presents an interesting and timely analysis of a wealth of transcriptome data upon immune stimulation. The unique combination of long-read Oxford Nanopore and short-read Illumina high-throughput sequencing across both human and mouse samples presents an opportunity many interesting inter-species immune response comparisons, as well as elucidation of full-length transcript information. This paper is well-written and has interesting validation and discussions regarding Aim2. My major concern is that the paper seems to narrow in on the characterization of Aim2 and class of RNA processing changes (alternative first exons) quite quickly without really delving into the rest of the data and how they arrived there. Below are my major comments and suggestions:1. I would have liked the authors to provide more insight into how they honed-in on specifically talking about first exon changes, by discussing more of the other RNA processing changes they found. There is cursory mention in the text and figures of other alternative exon or splice site changes. Firstly, other studies (including those referenced by the authors) have found hundreds RNA processing changes genome-wide upon immune stimulation – especially of cassette exons, alternative splice sites, and last exon/3'UTR changes. However here, the authors only find tens of changes (Figure 1B). Are they underpowered to identify changes and can they do any sort of analyses to show that they are sufficiently powered (# of sequencing reads and junctions, complexity of reads, etc)?

Our lab was interested in the global alternative splicing profiles of primary macrophages during inflammation. To determine this, we utilized both primary human and mouse macrophages and analyzed them using both JuncBASE and DRIMSeq to determine the significant differential splicing events. It was clear that in both species Alternative First Exon (AFE) changes are the most prevalent event-type of the significant events. To ensure that our AFE signature is inflammatory specific, we reanalyzed the datasets generated by Pai et al. using methods mirroring those we used for analyzing our own data. Following this, in supplemental figure 4 A-D we see that the results show a higher prevalence of AFE events using either adjusted p-value thresholds of.05 or.25 (PMID: 27690314). On the other hand, in the Pai publication, their figure 2A and C shows the AFE exons are prevalent amongst the significant events in both experimental conditions, even more prevalent than cassette exons (PMID: 27690314). (Manuscript Lines 110-114).

While AFE events are not the most prevalent amongst all event types in their analysis, this may be due to other factors such as difference in methodologies applied. We focused on alternative isoform changes that are coupled with alternative splice site usage; therefore, we did not examine TandemUTR events. To define alternative splicing events, we used JuncBASE which has features that allows for defining alternative first exons with higher confidence (PMID: 26294686). For statistical testing, we used a t-test through the JuncBASE package or DRIMSeq, which applies a Dirichlet multinomial framework. On the other hand, Pai et al., evaluated changes in RNA processing events with MISO, which uses Bayesian inference in order to detect differentially expressed isoforms. MISO identifies differentially expressed isoforms based on annotation, while JuncBASE allows for the detection of unannotated splicing changes. In addition, significance was defined differently between both studies. In our method, significance was generally defined as AS events with a corrected p-value <= 0.25 and |ΔPSI| >= 10%. In Pai et al., significance is designated to events following infection were defined as those with at least at least 10% of individuals having a Bayes Factor (BF) > = 5 and the | mean ΔPSI| ≥ 5%. The smaller threshold of | mean ΔPSI| ≥ 5% attributes to the increase in the number of splicing changes reported in Pai et al. compared to those reported in our study.

2. Similarly, I would also be interested in seeing an analysis indicating whether the 50 AFE events that overlap between the long-read and short-read sequencing analyses is a statistically significant overlap. Particularly, how many overlapping events would be expected given the difference in quantification power between the two methods?

To address this comment, we have revised the manuscript (Manuscript Lines 133-147, 324-334) to more clearly indicate that an alternative splicing identification and quantification analysis was not completed using the long-read sequencing data. The long-read sequencing data was only used to assemble high confidence isoforms to supplement the existing gencode transcript annotation to more accurately identify alternative splicing events in the Mouse data with JuncBASE and DRIMSeq. Figure supplement 5 indicates overlap of significant alternative splicing events found using either the gencode annotation file or our ‘Gencode + de novo’ (Manuscript Lines 115-117).

How many real AFE differences might the authors be missing because the long-read sequencing methods often do not have the power to identify them (ie. lower expressed genes in one or the other condition, thus dropout of isoforms and perhaps fewer isoform differences for differentially expressed genes).

For our primary BMDM direct RNA sequencing, we had an average read depth of 1 million reads per condition (+/- LPS). In a previous study of direct RNA sequencing, we found that increased sequencing depth will result in additional isoforms detected; therefore, we expect to see additional unannotated AFE with increased sequencing depth (PMID: 31740818). We have made sure to include this point in our Discussion. (Line 125-136, 327-333).

3. Second, for the non-AFE changes that they did find, there is very little discussion about what those changes might represent. Specifically: (a) how many changes are validated with long-read data?

To address this point, we have now generated venn diagrams for all other splicing events (SFigure 7). The venn diagrams show the overlap of differential alternative splicing events from short-read RNA-seq data using either the publicly available gencode.gtf file or our combined gencode and de novo.gtf file generated from our direct RNA nanopore sequencing data. (Manuscript Lines 132-141, 333-334).

(b) Is there any insight into specific domains being included/changed, especially using the long-read data?

Looking deeper in the motifs of the mouse AFE genes, in SFigure 12 (line 232-234, 373-375), we see that overall there are more motifs in LPS (inclusion) first exons in comparison to the CTL (exclusion) first exons. Musashi binding element, MBE, was the most prevalent motif identified in the first exons. Because AFE events were the predominant type of alternative isoform usage, we focused on motif analysis for AFE events.

(c) How many of these non-AFE changes overlap between species?

We generated 8 new venn diagrams to identify the overlap between mouse and human alternative splicing events during acute inflammation (SFigure 5). Interestingly, we see that alternative first exons have the most conservation of all other splicing events. (Manuscript Lines 115-117)

(d) Which types of genes show higher overlap between species and what are their characteristics (binding sites, etc)? To my knowledge, this is the first study that is really designed to properly really look at the conservation of splicing or RNA processing changes after immune activation, so I would love to see more analysis and discussion of this aspect genome-wide.

We appreciate this inquiry, and hope to take a deeper look into this on our next project. The scope of this study was primarily focused on alternative first exons since they are the most prevalent event type across species after inflammatory activation. Our first step to take a global approach in assessing the exclusion and inclusion of AFE was to look at length. It is known that longer 5’UTRs lead to a decrease in translational efficiency of a transcript (PMID: 11897027); therefore we looked at the length of all inclusion and exclusion exons (SFigure 12A). There is no significant difference between the length of inclusion and exclusion exons. However, it is important to see that the average ΔAFE was 37, meaning there is a slight shift in a longer 5’UTR in the inclusion AFEs. Furthermore, we took a deeper approach to investigate the possible motifs in the mouse inclusion (LPS) and exclusion (CTL) exons (SFigure 12C) by using RegRNA2.0. This is a web server that takes an unbiased approach to scan for 44 potential murine motifs. We show that there are more motifs identified in inclusion (LPS) exons, and the most prominent motif is the musashi binding element (MBE). (Manuscript Lines 232-234, 372374).

4. The authors define significant splicing changes as those with a p-value <= 0.25 and |dPSI| >= 10. I'd like some more clarification on whether this is an adjusted p-value (BH, FDR, or some other multiple test-corrected p-value). Especially if this is adjusted, I find it surprising that the authors are choosing such a liberal statistical confidence level and that even with such a liberal threshold, they are only getting tens of significant events. I would like the authors to at least show these same trends across multiple p-value thresholds or with rank threshold analysis (top 5%, top 10%, top 20%) to show biological trends.

Adjusted p-values from JuncBASE are adjusted using Benjamini and Hochberg (BH) by implementing the option --mt_correction BH. The adjusted p-values from DRIMSeq are also adjusted using Benjamini and Hochberg (BH) by default (Nowicka, M, et al.). A 0.05 significance threshold is somewhat of an arbitrary significance threshold that has been designated in the field and an adjusted p-value < 0.25 has been used in a variety of other studies in genomics (e.g. PMID: 30124010, PMID:27818134, PMID: 16199517) We felt that a less stringent threshold of 0.25 would be appropriate to use in this study as the identification of alternative splicing events following LPS stimulation began as an exploratory analysis and the implementation of a 0.05 adjusted p-value threshold yielded few results. As requested by the reviewer we performed a rank threshold analysis of the top 5%, 10%, 15%, 20%, and 25% based on uncorrected p-values and they show that alternative first exon events remains amongst the most prevalent significant events (SFigure 2-3). (Manuscript Lines 107-114).

(5) The authors introduce their long-read sequencing data by mentioning that they wanted to identify "additional splicing events that are not captured using short-read sequencing." They then go onto to only talk about novel first exon events identified with the long-read sequencing data. Did they identify any other non-AFE events in using the long-read that could then be quantified with the short read data?

Thank you for asking this question. This is a very critical point, and we hope our corrections make this point clearer throughout the manuscript. With our study we first did a differential splicing analysis, using the publicly available gencode gtf file. We took this a step further, by then performing long-read direct RNA sequencing to generate a new macrophage specific gtf file using FLAIR. We then reanalyzed our short-read RNAseq data with our de novo transcriptome. From this new transcriptome file, we did find other differential splicing events that were novel to macrophages, as well as events that were identified in the short read sequencing that were supported by our new gft file. This data can be found in both Figure 1C, as well as in the venn diagrams of supplemental figure 7.

And second, how do they quantify confidence for novel AFE isoforms, when long-read data seems to have lots of issues with properly sequencing the terminal ends of transcripts (particularly the 5' end when polyA primed, as occurs in ONT DirectRNA sequencing)? They mention the use of ATAC-seq data to show putative promoter support, but mention at one point in their methods that ATAC regions within 10kb of AFEs are considered.

FLAIR collapse, the software tool used in order to assemble high confidence isoforms from nanopore reads, is designed to handle truncated reads (Tang, AD., et al. 2020). In addition, we implemented an optional parameter of FLAIR collapse (-p) which instructs the tool to use an input file of coordinates (a BED file) corresponding to promoter regions to aid with distinguishing isoform start sites. When implementing this parameter, we only used promoter regions that were also supported by ATAC-seq data. The ATAC regions within 10kb of AFEs are considered for a separate analysis looking into chromatin accessibility. Methods found in the manuscript lines 786-793.

These seems like it could be a rather large region to be sure that the ATAC peak is specific to a novel AFE – what is the average distance between AFEs? Finally, I would love to also see the incorporation of CAGE-seq data (or other 5'end data) to validate the specific AFEs sites – which I believe the FANTOM consortium has across many human and mouse tissues.

We appreciate the inquiry. To address this comment, we elected to use the

FANTOM5 CAGE data and complete two analyses. In both analyses, “known” TSS are those that overlap TSS in the Gencode M18 annotation and “novel” TSS are those that do not. In the first analysis, we plotted the distribution of CAGE scores of the CAGE peaks that overlap known and novel TSS. The similarity in the distribution between the known and novel CAGE score distribution indicates to us that the novel TSS identified using Nanopore sequencing coupled with the bioinformatics tool, FLAIR, are truly TSS. A second analysis to further support that claim was completed in order to observe the proportion of known and novel TSS that overlap CAGE data corresponding to “true” TSS as classified with machine learning methods (PMID: 21596820). 45% of the novel and ~62% of known TSS overlap with CAGE peaks corresponding to “true” TSS, respectively. Because FANTOM5 CAGE data is an aggregation of data collected from multiple studies using different cell-types and experimental conditions, we expect that many of the non-overlapping CAGE peaks classified as “true” TSS are a result of these experimental differences. (Main text: 141-153, 333, Methods: 825-842).

Reviewer #4:In this manuscript, Robinson et al., identified alternative first exon (AFE) switching events conserved between mouse and human following macrophage inflammation. Using short and long-read sequencing, the authors identified a few unannotated transcription initiation sites (TSS) that are specific of an inflammatory response. Among those, they centered on an unannotated TSS in the Aim2 gene that drives expression of a novel isoform regulated by an iron-responsive element in its 5′UTR.While previous work had documented crucial AFE switching events in many other biological contexts, Robinson et al. presents here an interesting AFE switching event that can have potential implications for our understanding of the molecular regulation of the innate immune response. For publication in eLife, I would expect further progress on global mechanisms and biological relevance of these AFE switching events, as well as evidence that the AFE are truly first exons/TSSs.1. Are the AFEs truly first exons/TSS? While both short-read and long-read sequencing detected changes in alternative splicing choices, neither of those are optimal methodologies to analyze first exons. Therefore, I suggest to use a more specialized method to identify (and quantify) more accurately the usage of first exons. Globally, cap analysis of gene expression (CAGE) would be ideal. For validation of specific AFE changes, the qPCR technique has a few issues. First, it does not have nucleotide resolution, so the authors should not refer to TSSs if they used this technique for validation.

We appreciate the inquiry about the confidence of our isoforms that have novel TSS. To address this comment, we elected to use the FANTOM5 CAGE data and complete two analyses. In both analyses, “known” TSS are those that overlap TSS in the Gencode M18 annotation and “novel” TSS are those that do not. In the first analysis, we plotted the distribution of CAGE scores of the CAGE peaks that overlap known and novel TSS. The similarity in the distribution between the known and novel CAGE score distribution indicates to us that the novel TSS identified using Nanopore sequencing coupled with the bioinformatics tool, FLAIR, are truly TSS. A second analysis to further support that claim was completed in order to observe the proportion of known and novel TSS that overlap CAGE data corresponding to “true” TSS as classified with machine learning methods (PMID: 21596820). 45% of the novel and ~62% of known TSS overlap with CAGE peaks corresponding to “true” TSS, respectively. Because FANTOM5 CAGE data is an aggregation of data collected from multiple studies using different cell-types and experimental conditions, we expect that many of the non-overlapping CAGE peaks classified as “true” TSS are a result of these experimental differences. (Manuscript Lines 141-153, 333, Methods: 825-842).

Second, many downstream first exons are also used as internal exons in other isoforms. There is not a direct technology to analyze specifically first exons/TSSs here. Also, RNA-sequencing technologies, depending on their depth, can definitely miss specific isoforms. Considering a low coverage in 5'end of genes in RNA-seq analysis, this is particularly important for first exons. A qPCR would only analyze the well-known TSSs. Thus, 5'RACE or a similar technology should be perform to assess the relative usage of AFE specifically.

While it is true that there are limitations of these RNA sequencing technologies to determine the true first nucleotide, we still are able to determine nearly all splice junctions depending on read depth. Our illumina sequencing run was very deep, around 30million per library. Our nanopore sequencing libraries were much lower, at 1million reads per library, but they were used to generate a new transcriptome using FLAIR. The limitations of nanopore, as discovered by Mulroney et al. show that direct-RNA long read nanopore sequencing is unable to detect the first ~11nt of a transcript (DOI: 10.1101/2020.11.18.389049). Then we re-analyzed our illumina RNA-sequencing libraries using our new transcriptome. Thus the exact transcriptional start site is not critical in the definition of alternative first exon. We also used orthogonal ATAC-seq data to provide additional confidence that the first exon was at a putative promoter.

2. Global mechanism. The authors assumed that the mechanism of AFE switching is generated by transcription initiation and looked for transcription factors binding and chromatin structure modifications in promoters. However, they did not ruled out the possibility that the global switching effect is a post-transcriptional regulation, such as differential mRNA stability. A transcription initiation measurement (e.g., 4SU metabolic labelling) is necessary to demonstrate that the changes in AFE usage are co-transcriptional.

Thank you for this astute point. While we did not assess differential mRNA stability of the identified isoforms, we can confirm many of the events by using our ATAC-seq and ChIP-seq analysis. For instance, by examining the p65 and IRF3 transcription factors, we were able to show that 37 of the promoters are regulated by either p65 or IRF3. Additionally, 26 promoters of the AFE genes undergo chromatic accessibility changes during LPS stimulation (Figure 3). While 47 of the 95 AFE events are unaccounted for, this can be explained by our global assessment of promoters of the AFE events (Figure 3B), showing that many of the transcription factors enriched are regulated by metabolism and other biological processes (Line 203-205, 353-355). Assessing additional mechanisms that may be driving these AFE events we fell are outside the scope of this current study.

In addition, in terms of their ATAC-Seq analysis, the chromatin structure changes in promoters can be cause or consequence of transcription initiation. Thus, it should not be listed as one mechanism driving the expression of AFE events (line 145). Also, to demonstrate a mechanism based on transcription factor binding more than 2 transcription factors should be considered. In any case, the expression patterns of the transcription factors considered are not clear.

Thank you for this astute point, we have changed the text to address this concern. We investigated transcription factor binding as another potential contributing factor to the changes in AFE usage. The transcription factors we chose, p65 and IRF3, are two of the key inflammatory transcription factors driving thousands of immune response genes (described line 4346). Additionally, there are not many ChIP-seq datasets available for transcription factors of interest in primary BMDMs. The datasets chosen are for an inflammatory time course, 0-120min, with 120min chosen for analysis. This is the time point in which the highest binding of IRF3 or p65 is seen. While additional examination of other TFs could provide more mechanistic understanding we feel those studies are outside the scope of this paper. However, as an alternative approach to analyzing more ChIP-seq data from primary BMDMs, we performed a HOMER analysis of all AFE promoters (Figure 3B). We found that the majority of TFs are regulated by metabolism (Manuscript Lines 203-205, 353-355).

As a minor note, the bioinformatic analysis of the two promoters regions driving the isoforms of Aim2 (line 156) is not explained in the method section.

Thank you for this point, we have edited our methods to include the promoter analysis on Aim2 (Manuscript Lines 892-899).

3. Biological relevance. Could the authors evaluate whether the translation regulation of Aim2 based on its AFE switching is a more generalized phenomenon?

The general phenomenon of AFE usage has been seen previously in a handful of plant or *Drosophila* studies, but in the context of inflammation in mammalian cells is a novel finding (PMID: 17941993, PMID: 16344560, PMID: 19178699). For our results within this manuscript, we can definitively say that the AFE of Aim2 downregulates the protein expression through an IRE motif by our GFP expression experiments, polysome profiling experiments, western blot analysis of primary BMDMs and our Cas9 IRE KO experiments in immortalized BMDMs (Figure 4 and SFigure 15). (Line 374-389).

Furthermore, we took a deeper approach to investigate the possible motifs in the mouse inclusion (LPS) and exclusion (CTL) exons (SFigure 12C) by using RegRNA2.0. This is a web server that takes an unbiased approach to scan for 44 potential murine motifs. We show that there are more motifs identified in inclusion (LPS) exons, and the most prominent motif is the musashi binding element (MBE). (Line 232-234, 372-374) IRE motifs were not found in any other AFE event.

Are there any global gene regulation changes triggered by the other genes with significant changes is AFE usage?

After performing a global assessment of all inclusion and exclusion AFEs, using RegRNA2.0, we found that the inclusion AFE events had more motifs than the exclusion exons (SFigure 12, line 232-234). This could mean that there are additional post-transcriptional regulations affecting the translation of the genes during inflammation. It was beyond the scope of the paper to do another global sequencing experiment to assess gene regulation of the isoforms, such as polysome profiling or ribosome profiling. While our study does not look at the global translational implication of all AFE events, it does unearth a novel mechanism of an AFE event in the Aim2 protein coding gene. Aim2 was originally identified in a seminal 1997 study (PMID: 9242382) as an interferon inducible gene. From our work we now know that it is this alternative isoform that is inducible and it leads to a decrease in protein expression most likely as a way to control the pathway and return it to homeostasis.